# IMAGE REGISTRATION IS A GEOMETRIC DEEP LEARNING TASK

## ABSTRACT

Data-driven deformable image registration methods predominantly rely on operations that process grid-like inputs. However, applying deformable transformations to an image results in a warped space that deviates from a rigid grid structure. Consequently, data-driven approaches with sequential deformations have to apply grid resampling operations between each deformation step. While artifacts caused by resampling are negligible in high-resolution images, the resampling of sparse, high-dimensional feature grids introduces errors that affect the deformation modeling process. Taking inspiration from Lagrangian reference frames of deformation fields, our work introduces a novel paradigm for data-driven deformable image registration that utilizes geometric deep-learning principles to model deformations without grid requirements. Specifically, we model image features as a set of nodes that freely move in Euclidean space, update their coordinates under graph operations, and dynamically readjust their local neighborhoods. We employ this formulation to construct a multi-resolution deformable registration model, where deformation layers iteratively refine the overall transformation at each resolution without intermediate resampling operations on the feature grids. We investigate our method's ability to fully deformably capture large deformations across a number of medical imaging registration tasks. In particular, we apply our approach (GeoReg) to the registration of inter-subject brain MR images and inhale-exhale lung CT images, showing on par performance with the current state-of-the-art methods. We believe our contribution open up avenues of research to reduce the black-box nature of current learned registration paradigms by explicitly modeling the transformation within the architecture.

## 1 INTRODUCTION

Image registration is an indispensable tool in medical image analysis that aligns anatomically or functionally corresponding regions across images, often from different modalities and time points (Sotiras et al., 2013). In particular, deformable registration aims to estimate a non-linear transformation that maps the *source* image to the coordinate space of the *target* image. Since the advent of deep learning, data-driven methods have been proposed (Haskins et al., 2020; Xiao et al., 2021) to leverage learned transformation priors over an image cohort, reducing the search space of plausible transformations.

**Single-stream approaches.** Since images are usually represented as grids of pixels, data-driven approaches typically employ convolutional kernels to model transformations. Early works have commonly adopted a simplistic approach of concatenating the source and target images as input channels to a convolutional network (Balakrishnan et al., 2019; Dalca et al., 2018; Mok & Chung, 2020a; Qiu et al., 2021; Zhao et al., 2019). However, this concatenation produces different inputs to a network on every possible misalignment of the counterpart image. This subsequently causes the feature extraction to learn distinct representations throughout the network on each possible misalignment, increasing the task's complexity.

A common necessary preprocessing technique employed to mitigate this issue involves an exhaustive search for an initial affine alignment. This reduces the degrees of freedom in the transformation parameters by guaranteeing that similar features are captured in a consistent spatial context, thus reducing the range of representations experienced by the network. Recent works combat the misalignment-

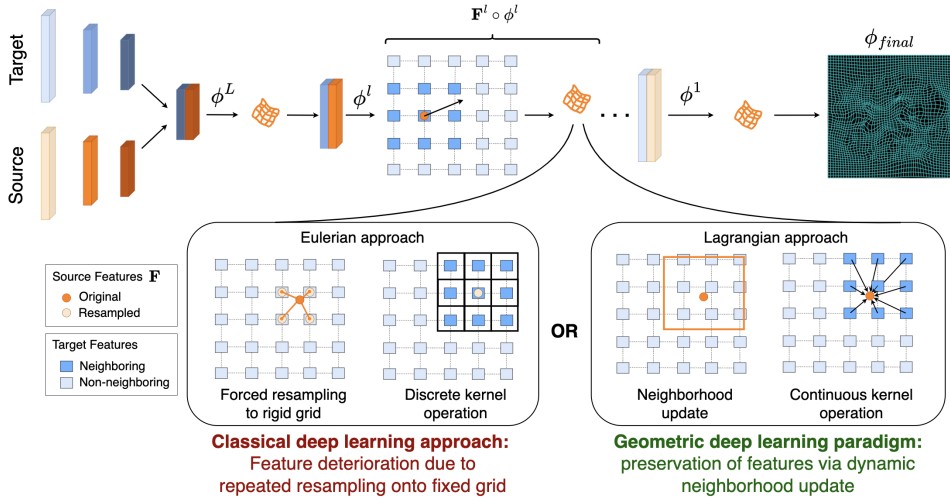

Figure 1: A classical deep learning setting operates on an Eulerian framework, where an implicit grid is required to predict a transformation. After a deformation is applied, this forces resampling to perform subsequent predictions. Our approach employs geometric deep learning to formulate the registration task as a grid-independent process using a Lagrangian reference frame.

dependent complexity by incorporating transformer layers throughout the network (Chen et al., 2022; 2023; Liu et al., 2022; Meng et al., 2022; Wang et al., 2023; Zhu & Lu, 2022). This enables greater flexibility in the feature extraction process as the transformer layer's attention mechanism is able to establish non-local spatial relationships at the cost of increased learnable parameters. Similarly, cascading approaches have shown increased accuracy by recovering the misalignment progressively, modeling the transformation as a sequence of deformations (Hu et al., 2022; Sandkühler et al., 2019; Zhao et al., 2019).

**Dual-stream approaches.** A commonly adopted technique to avoid concatenating the images at the input is using dual-stream encoders (Hu et al., 2022; Wang et al., 2023; Kang et al., 2022; Liu et al., 2022; Meng et al., 2022). This approach utilizes two separate encoders that individually extract features from source and target images, allowing for misalignment-independent representations throughout most of the network. Moreover, multi-resolution methods (Hu et al., 2022; Kang et al., 2022; Meng et al., 2022; Mok & Chung, 2020a; Wang et al., 2023) estimate the transformation at multiple levels during the decoding process in a coarse-to-fine fashion. This proves to be a strong architectural prior towards capturing large transformations in a parameter-efficient manner.

Despite their effectiveness, the layers of these architectures remain strictly confined to grid-structured features due to their dependence on discrete kernel operations. Consequently, as warping operations are applied to the feature grids in the decoding process, the space needs to be resampled to adhere to the fixed grid positions (Figure 1). Although negligible on the pixel level, resampling operations become increasingly inaccurate on sparse, high-dimensional feature spaces. This issue, termed *the curse of dimensionality*, causes naive interpolation to deteriorate the quality of features being propagated to later sections of the prediction process Verleysen & François (2005).

**Reference frames.** Current data-driven registration methods predominantly rely on an Eulerian (Batchelor, 1967) frame of reference for modeling deformation fields, whereby the deformation field is observed at specific locations in space. Conversely, the Lagrangian (Batchelor, 1967) specification is an alternative choice of modeling motion that tracks the positions of infinitesimal parcels (Brun et al., 2010; Lester et al., 1999; Thirion, 1998; Vercauteren et al., 2009). These concepts carry striking similarities with recent research in the field of geometric deep learning, where the motion of sparse point-like objects is modeled using learned functions (Farahani & Hamker, 2022; Fuchs et al., 2020; Kashefi et al., 2021). This design choice obviates the aforementioned resampling issues by not being confined to a grid-based frame of reference, removing the costly memory requirement of tracking an arbitrarily precise voxelized volume. While geometric deep learning has been used in certain contexts for point cloud and cortical surface registration (Hansen & Heinrich, 2021a;b; Suliman et al.,

2022; Shen et al., 2021; Hoopes et al., 2022), to the best of our knowledge, no previous work offers a framing of *deformable* image registration within the geometric deep learning paradigm.

**Contributions.** In this work, we propose a novel paradigm for data-driven image registration by viewing the deformation modeling process through the lens of geometric deep learning. We formulate the task as a multi-scale process of deformation operations, where feature grids are modeled under a Lagrangian framing of free-floating nodes influenced by neighborhood interactions. Unlike existing approaches, our method explicitly models coordinates and features independently, performing node-wise operations using continuous learnable kernels. This formulation enables us to completely avoid grid-based constraints on inter-node structure, removing the requirement for intermediate warping operations between transformations.

Our contributions can be summarized as follows:

- We establish a mathematical foundation to formulate deformable registration in a continuous domain, avoiding the need for interpolation in the feature space. We achieve this under a Lagrangian reference frame utilizing the geometric deep-learning paradigm.

- We propose a data-driven form of interpolation demonstrating local support, which facilitates multi-scale deformation modeling by learning to propagate deformations across resolutions.

- We demonstrate the effectiveness of our formulation by reporting improvements over current state-of-the-art deformable registration in the context of medical imaging and showing our model's ability to recover large deformations. We make our code publicly available[1].

## 2 METHOD

In this section, we first formally establish the limitations imposed on deformable image registration by the grid constraints of Eulerian frameworks. Afterwards, we establish a Lagrangian formulation that does not make any grid assumptions (section 2.1). Within this context, we highlight the advantages offered by geometric deep learning in modeling deformations as interactions between free-floating features (section 2.2). Next, we propose a data-driven form of local interpolation, which facilitates multi-scale deformation modeling by learning to propagate deformations across resolutions (section 2.3). Finally, we combine these ideas to construct an end-to-end trainable neural network capable of learning deformable registration in continuous domains in a coarse-to-fine fashion (section 2.4).

### 2.1 DEFORMATION WITHOUT GRID CONSTRAINTS

An image $I$ can be interpreted as a finite grid of measurements $I \in \mathbb{R}^{H \times W \times D}$ which represents a discrete subset of a continuous domain $\Omega$, and where $H, W, D$ are the spatial dimensions of the voxel grid in the 3-dimensional case.

Given a target $T$ and a source $S$ image, deformable image registration aims to find an optimal spatial deformation field $\phi^* = \arg\min_\phi \mathcal{J}(T, S, \phi)$, with $\phi : \mathbb{R}^n \to \mathbb{R}^n$, such that the transformed source image $S \circ \phi$ is most similar to the target image $T$. As such, the overall objective $\mathcal{J}$ is defined as:

$$\mathcal{J}(T, S, \phi) = \mathcal{D}(T, S \circ \phi) + \lambda \mathcal{R}(\phi), \tag{1}$$

where $\mathcal{D} : \Omega \times \Omega \to \mathbb{R}^n$ is an image dissimilarity measure responsible for driving the deformation, and $\mathcal{R}$ is a smoothness regularization on the transformation whose magnitude is weighted by $\lambda$.

Data-driven methods implement learnable functions responsible for modeling a deformable transformation $\phi$ as a neural network $\tau_\theta(T, S)$ parametrized by a set of learnable weights $\theta$. These approaches model the transformation based on spatial features $\boldsymbol{F}^S$ and $\boldsymbol{F}^T$ extracted from $S$ and $T$.

Images are usually represented as grids of nodes containing pixel information. By making relative positions between neighboring nodes constant across the structure, grid representations carry implicit assumptions about the homogeneity of the structure. As such, attempting to construct a neural function $\tau$ by learning grid-based operations (such as convolutions with discrete kernels) on the

---

[1]https://anonymous.4open.science/r/GeoReg-D567

features $\boldsymbol{F}$ of a given image may appear enticing. However, these grid-reliant architectures generally confine the transformation prediction to a grid, resulting in an Eulerian framework whereby the deformation field is only defined on the specific locations in space defined by the grid. As spatial transformations are applied to the source domain $S$, the rigid assumption on grid structure requires a resampling of the feature space to estimate the feature values at the grid positions, which may result in deteriorated features.

Instead of relying on grid representations, we propose to explicitly represent the source domain under a Lagrangian framework via an unstructured set of feature-coordinate tuples $S = (\boldsymbol{F}^S, \boldsymbol{X}^S)$, where $\boldsymbol{F}^S \in \mathbb{R}^{|S| \times d}, \boldsymbol{X}^S \in \mathbb{R}^{|S| \times n}$. Now, each node $s \in S$ can be regarded as a discrete observation of a feature $\mathbf{f}^s$ embedded in Euclidean space at coordinate $\mathbf{x}^s$. Here, the feature component $\mathbf{f}^s$ of a given node $s$ may represent anything from gray-scale intensities ($d = 1$) to higher dimensional feature descriptors such as those extracted by a feature encoder. This formulation allows us to model the deformation process as a function $\tau$ acting on the coordinate component $\mathbf{x}^s$ of each $s \in S$ without requiring the feature $\mathbf{f}^s$ to be modified. This Lagrangian framework is especially enticing to coarse, high-dimensional feature spaces that would otherwise suffer from the curse of dimensionality under interpolation operations (Bronstein et al., 2021).

## 2.2 Deformation modeling function $\tau$

Formulating the source domain $S$ as a set of nodes with real-valued coordinates, with no necessity of adhering to a rigid grid, requires a generalized form of learned functions that can handle continuous domains. Precisely, we need to define a deformation function $\tau(\cdot)$ that is able to act node-wise on the set of source nodes $s \in S$, while being capable of handling neighborhoods of target nodes $T_{\mathcal{N}_s} \subset T$ with arbitrary real-valued coordinates relative to a given $s$. This formulation naturally leads us to the realm of geometric deep learning, where graph neural networks are leveraged to model learned functions on geometric graph structures.

We begin by using the previously introduced notation of feature-coordinate pairs for the nodes of the target domain $T = (\boldsymbol{F}^T, \boldsymbol{X}^T)$. Using the source $S$ and target $T$ sets, we define the domain of the deformation function $\tau$ as a directional graph $\mathcal{G}^\tau = (\boldsymbol{A}^\tau, [\boldsymbol{F}^T, \boldsymbol{F}^S], [\boldsymbol{X}^T, \boldsymbol{X}^S])$ describing how each of the source nodes $s \in S$ interacts with the target domain $T$. The adjacency matrix $\boldsymbol{A}^\tau \in \{0, 1\}$ describes the presence or absence of edges between all node pairs. In practice, since we only want to model how a given source node $s$ should deform given its local neighborhood in the target domain, we only need to model one quadrant of the full adjacency matrix $\boldsymbol{A}^\tau \in \{0, 1\}^{T \times S}$. A row in $\boldsymbol{A}^\tau$ represents the neighborhood $T_{\mathcal{N}_s}$ of a source node $s$ within the target domain $T$ by defining which subset of target nodes contain directional edges to $s$, such that $T_{\mathcal{N}_s} = \{t \mid \forall t \in \boldsymbol{A}^\tau_{[s]}, \boldsymbol{A}^\tau_{[s,t]} = 1\}$. The values of a row $\boldsymbol{A}^\tau_{[s]}$ are computed using k-nearest neighbors based on $|\mathbf{x}^t - \mathbf{x}^s|$ node distances.

Given a graph $\mathcal{G}^\tau$, we can define a generalized framing of a convolution by modeling a learnable kernel as a continuous function $\psi$. This formulation can operate on both grid and non-grid layouts alike. When centered on a node $s$, the function $\psi$ computes an activation to a set of neighboring target nodes $T_{\mathcal{N}_s}$ based on their features $\mathbf{f}^t$ and relative coordinates to the source node $(\mathbf{x}^t - \mathbf{x}^s)$:

$$\mathbf{f}' = \frac{1}{|T_{\mathcal{N}_s}|} \sum_{t \in T_{\mathcal{N}_s}} \psi \left( \mathbf{f}^t, \mathbf{f}^s, \left( \mathbf{x}^t - \mathbf{x}^s \right) \right) \tag{2}$$

The function $\psi$ is typically implemented as a learnable linear projection with a non-linearity. The output feature of this convolution operation can then be further projected into the vector of size $n$ to predict a deformation $\phi \in \mathbb{R}^n$ for that given node. This establishes a basis for learning a continuous deformation model $\tau(\mathcal{G}^\tau)$ that can be applied to a continuous domain in order to predict deformations $\phi$ for any node $s$.

This process can be iteratively performed, creating a chain of deformations $\boldsymbol{X}^S \circ \tau(\mathcal{G}^\tau_1) \circ ... \circ \tau(\mathcal{G}^\tau_N)$ that refine the transformation prediction in a cascading fashion. As the source nodes are transformed relative to $T$, the graph $\mathcal{G}^\tau$ can simply be recomputed using the newly transformed coordinates (see Figure 2 (iv)). Unlike other state-of-the-art cascading approaches, our grid-independent formulation requires no intermediate resampling-to-a-grid operations, maintaining feature integrity along the deformation chain. We refer the reader to the pseudocode in Appendix B1 for a detailed overview.

## 2.3 MULTI-RESOLUTION INTERPOLATION FUNCTION $\boldsymbol{\delta}$

The optimization process of real-world registration tasks is highly non-convex. A prevalent strategy in literature to overcome local minima when dealing with complex transformations is the usage of a multi-resolution strategy. In line with this established paradigm, we explore the application of $\tau$ deformation functions across entire feature pyramids in coarse-to-fine multi-resolution settings. In this section, we aim to define a function $\delta(\cdot)$ capable of chaining deformations in a coarse-to-fine fashion across resolutions without feature interpolation.

### 2.3.1 LOCALLY-WEIGHTED INTERPOLATION

A commonly adopted technique in parametric image registration involves predicting deformations at a coarse spacing and interpolating to the desired resolution via continuous mapping functions. Generally, these mappings are formulated in the context of a set of control points $C$ exerting influences on the interpolation at a given point in space via local basis functions. Particularly in the case of b-spline basis functions, the interpolation process exhibits the property of local support, implying that a small, localized change has a restricted impact and does not influence the entire domain.

The transformation $\phi$ at an arbitrary point in space $i$ with coordinates $\mathbf{x}^i$ is the resulting interpolation of the transformation values $\phi^c$ of its neighboring control points $c \in \mathcal{N}^C$. This interpolation is weighted using basis functions $v$, based on relative positions between the given point $i$ and each control point $c$:

$$\phi(\mathbf{x}^i) = \sum_{c \in C_{\mathcal{N}_i}} \overbrace{v\left(\mathbf{x}^c - \mathbf{x}^i\right)}^{\text{weight coefficient}} \phi^c \tag{3}$$

This concept of locally weighting a transformation, based on relative location to control points, serves as a powerful heuristic for introducing local support. However, we argue that making the interpolation mechanism aware of image features is the key to building improved interpolation functions.

### 2.3.2 CROSS-ATTENTION AS DATA-DRIVEN INTERPOLATION

The attention mechanism has been applied to illustrate a more general version of the convolution operation (Bronstein et al., 2021) described in Eq. (2). In the context of registration, graph convolutions already display desired local properties by restricting message-passing within local neighborhoods. The attention operation extends this principle by dynamically "masking out" irrelevant neighboring nodes. Unlike the convolution's simple uniformly-weighted aggregation of neighbors' responses, the attention mechanism allows a node to compute a form of learned weighted averaging based on its neighbors' features and relative positions.

$$\mathbf{f}' = \sum_{c \in C_{\mathcal{N}_i}} \overbrace{a(\mathbf{f}^c, \mathbf{f}^i, \left(\mathbf{x}^c - \mathbf{x}^i\right))}^{\text{weight coefficient}} \psi\left(\mathbf{f}^c, \mathbf{f}^i, \left(\mathbf{x}^c - \mathbf{x}^i\right)\right) \tag{4}$$

where $C_{\mathcal{N}_i}$ is the neighborhood of control points to node $i$. When nodes $i$ and $c$ belong to different domains (e.g., different images or resolution levels), the operation described in Eq. (4) is referred to as *cross-attention*. Here, the attention function $a$ is constrained to be in the range $[0, 1]$ by applying a softmax operation over all neighboring control points such that $\sum_{c \in C_{\mathcal{N}_i}} a(\cdot) = 1$. For further implementation details, we refer the reader to section 2.4. The concept of attention as dynamically weighting neighboring nodes as outlined in Eq. (4) offers strong similarities to the principles of parametric registration methods outlined in Eq. (3). Similarly to how parametric interpolation uses a preset weighting function $v\left(\mathbf{x}^c - \mathbf{x}^i\right)$ on neighboring control points, local attention uses a learned weighting function $a(\mathbf{f}^i, \mathbf{f}^c, \left(\mathbf{x}^c - \mathbf{x}^i\right))$. In the local attention setting, since a given node only interacts with its spatially restricted neighborhood (and not with the entire space), a localized change does not affect the entire domain, effectively offering properties of local support. The benefit of the attention mechanism here is the ability to condition the weighting coefficients not only on relative coordinates but also on the learned features present in the operation.

While not a direct form of deformation interpolation, local cross-attention offers a way to refine a node's location given its current neighborhood of control points. By placing a node near a set of

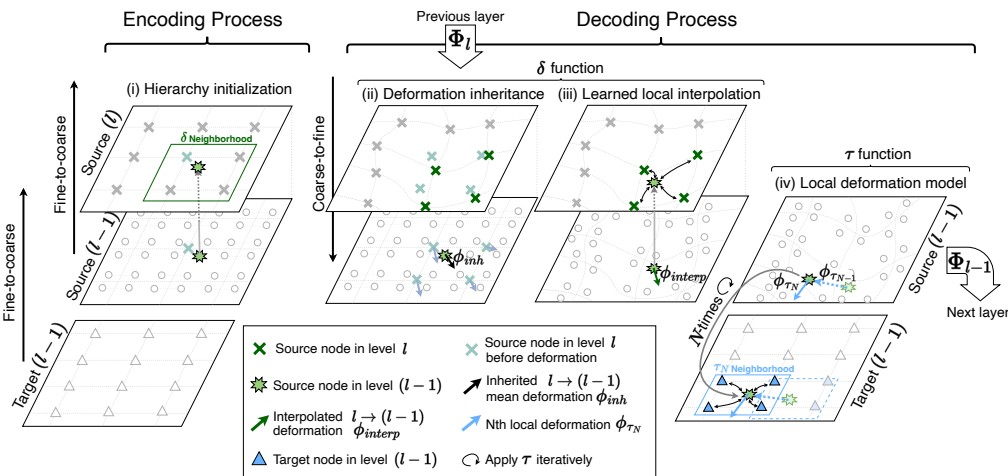

Figure 2: Architectural outline of a given resolution layer. (i) During feature extraction, pooling operations track cross-resolution hierarchies. (ii) During decoding, each child node inherits and further refines its transformation by performing local attention over its neighborhood of parents. (iii) Deformations are predicted by performing local attention over local target nodes. This operation can be iteratively performed an arbitrary number of times by dynamically updating a source node's current neighborhood in the target domain.

control points, the attention mechanism produces a weighted feature vector, which can be subsequently projected into the vector of size $n$ to predict a learned deformation $\phi \in \mathbb{R}^n$. This gives us a building block to define a local function $\delta(\cdot)$ that interpolates a deformation through local cross-attention on a continuous domain.

As depicted in Figure 2 (ii) & (iii), this component enables us to carry deformations across feature pyramid levels in the multi-scale decoding process, further circumventing the need for grid-based resampling operations.

### 2.3.3 HIERARCHY GRAPHS AS INTERPOLATION DOMAINS

Our aim is to define a hierarchical process by which a finer resolution level inherits and subsequently refines the transformation of a coarser level. As such, we refer to control nodes at level $l$ as *parents* and the nodes at the following level $l - 1$ as *children*.

To define the domain of the cross-resolution interpolation function $\delta(\cdot)$, we need to establish a graph representing parent-child connections. Concretely, given a set of control nodes $S^l$ at the $l$-th pyramid level and a set of children nodes $S^{l-1}$, we define a directional graph $\mathcal{G}^l = (\boldsymbol{A}^l, [\boldsymbol{F}^l, \boldsymbol{F}^{l-1}], [\boldsymbol{X}^l, \boldsymbol{X}^{l-1}])$. Here, $\boldsymbol{A}^l$ is a static matrix exclusively containing edges connecting children nodes to their k-nearest neighboring parent nodes.

**Hierarchy graph initialization.** As depicted in Figure 2 (i), we use the feature extraction process at the encoder to define parent-child hierarchies based on pooling operations. The feature extraction process establishes a hierarchy of children-parent nodes $S^{\{1,...,L\}}$, which all live in a combined Euclidean space across $L$ resolution levels. A parent node at level $l$ derived from a region of children nodes at level $l - 1$ has its position initialized at the center of the pooling window. This hierarchy establishes the cross-resolution neighborhoods that facilitate the interpolation operation of each given child node $s \in S^{l-1}$ during the decoding process.

**Transformation inheritance.** As depicted in Figure 2 (ii), an initial transformation is inherited by each child node at level $l - 1$ based on the average transformation of their neighborhoods at resolution $l$. This rough initial approximation significantly reduces the remaining transformation component left to be modeled by the feature-aware learned interpolation. We refer the reader to the pseudocode in Appendix B.2 for a detailed overview of this process.

## 2.4 IMPLEMENTATION DETAILS

**Image encoder.** Since our approach requires the source and target features to interact, it is necessary to implement a dual-stream encoder architecture. As a result, feature pyramids are extracted for the source $S$ and target $T$ images independently using the same set of weights. We refer the reader to Appendix A for a schematic of the overall model. The encoder consists of two convolutional residual blocks per resolution followed by pooling layers. The encoder is composed of 6 layers, each made up of two residual blocks, each with $[16, 32, 32, 64, 64, 128]$ channels. Average pooling $[2 \times 2 \times 2]$ operations are applied between each encoder layer.

**Deformation decoder.** The feature pyramid is then decoded in a coarse-to-fine fashion across the 6 resolutions. Across our experiments, we find that local deformations are sufficiently well-modeled at coarser resolutions in the decoder. Estimating the majority of the transformation at coarser levels reduces the registration burden of finer resolution layers, as only smaller local deformations are left to be recovered (see Appendix F for further detail). These insights allow us to refrain from applying $\tau$ at the finest layers where resolutions are the largest, massively decreasing our memory footprint.

**Memory efficient neighborhood computation.** While the methodology outlined in sections 2.2 and 2.3 are formalized using graph notation, the grid structure of our data allows us to design highly memory efficient implementations of $\tau$ and $\delta$ layers. Finding nearest neighbors in sparse data is the biggest memory bottleneck due to having to compute $O(N^2)$ distance calculations relative to the number of nodes $N$. Our convolutional encoder providing us with feature grids, lets us use grid-unfolding operations to find the nearest neighbors in a $(k_x, k_y, k_z)$ kernel around a central node. The $\tau$ function's neighborhood computations are performed by first unfolding the target domain into all its possible neighbourhoods. Then, we can index a source node's corresponding target neighborhood by mapping the current source coordinates into the index space of the target grid and rounding to the closest integer. Similarly, the $\delta$ function applies a repeated interleaving operation to upsample the parent unfolded neighborhoods into the same dimensions as the children grid. This is all implemented using standard built-in Pytorch functions that allow for efficient GPU parallelism. Wherever possible, we make use of pointers to the original data structures for minimal memory footprints. We refer readers to Appendix D for an overview on VRAM requirements of various registration baselines.

**Attention layers.** The attention mechanism described in Eq. (4) is implemented using standard transformer layers. The function takes the following matrix form:

$$\mathbf{f}' = \mathrm{softmax}\left( \frac{Q\left(\mathbf{f}^i\right) \cdot K(\mathbf{F}_{\mathcal{N}_i}, \left(\mathbf{X}_{\mathcal{N}_i} - \mathbf{x}^i\right))^\top}{\sqrt{d}} \right) V(\mathbf{F}_{\mathcal{N}_i}, \left(\mathbf{X}_{\mathcal{N}_i} - \mathbf{x}^i\right)) \tag{5}$$

Here, $V$ is the *value* matrix, playing an equivalent role to the function $\psi$ in Eq. (2) and (4). The attention scores are computed by taking the dot product *query* $Q$ and *key* $K$ vectors. We represent $Q$, $K$, and $V$ as functions to indicate the presence of positional embedding steps, whereby information about the relative locations of neighboring nodes is encoded into the node's feature vector. We use Fourier features ((Tancik et al., 2020)) as our choice of embedding function. The concept of an input set of nodes computing attention scores to a different set of nodes is referred to as *cross-attention*.

**Hyperparameters.** Tuning neighborhood sizes required striking a balance between receptive field width and memory limitations. For $\tau$, we selected neighborhoods of $5^3$ for coarser resolutions (to allow for wide receptive fields), while finer resolutions used $3^3$ neighborhoods to reduce memory footprint. Neighborhoods in $\delta$ always used the closest $3^3$ neighboring parent nodes. To calculate the loss at each resolution level, we employ normalized cross-correlation (NCC) as the dissimilarity metric. Furthermore, we utilize bending energy (Rueckert et al., 1999) as a regularizer to ensure a smooth final transformation at each resolution. The approach is trained end-to-end using the ADAM optimizer with a $10^{-4}$ learning rate for a maximum of 1000 epochs. Model training was carried out on an NVIDIA A40 GPU with 40GB VRAM over the course of 3 days. For further parameter details, we refer to our repository.

## 3 RESULTS AND DISCUSSION

### 3.1 DATASETS, BASELINES AND EVALUATION METRICS

We evaluate our work using the CamCAN T1w-T1w and T1w-T2w brain datasets (Shafto et al., 2014; Taylor et al., 2017) and the publicly available benchmark National Lung Screening Trial (NLST) dataset (team, 2011) from the Learn2Reg challenge 2022 (Hering et al., 2022). We refer the reader to section C of the Appendix for more information about the data, the pre-processing steps, the segmentation labels, and the key-point extraction.

We compare our method (GeoReg) against several conventional iterative methods and learning-based image registration models. Regarding the iterative optimization methods, we choose from the Medical Image Registration ToolKit (MIRTK) (Schuh et al., 2014), a widely-used free-form deformation (FFD) iterative optimization method that supports multi-resolution and parametric b-spline-based registration. Additionally, we compare against the widely adopted symmetric diffeomorphic alorithm SyN (Avants et al., 2008) from the ANTs (Avants et al., 2009) framework, as well as Large deformation diffeomorphic metric mapping (LDDMM (Beg et al., 2005)).Our learning-based baselines are comprised of Voxelmorph (Balakrishnan et al., 2019), a single-stage CNN, LapIRN (Mok & Chung, 2020b) a multi-resolution registration CNN that aims to capture large deformations in a coarse-to-fine manner, Transmorph (Chen et al., 2022) that uses a SwinTransformer-based encoder, Recursive Cascaded Networks (RCN) (Zhao et al., 2019) which estimates the deformation progressively using a cascading CNN architecture, the dual-stream pyramid registration network (D-PRNet) (Kang et al., 2022) that gradually refines the multi-level predicted deformation fields in a coarse-to-fine manner via sequential warping, and FourierNet (Jia et al., 2023) that learns a low-representation displacement filed in a band-limited Fourier domain and then uses a model-driven decoder to obtain the dense, full-resolution displacement field. To ablate the contribution of the proposed interpolation mechanism ($\delta$) on top of our multi-resolution $\tau$ design, we replace the proposed learned interpolation component ($\delta$) with bilinear feature warping. In the following, we denote this ablation baseline as "feat. warp".

The accuracy of the registration is evaluated using the segmentation metrics Dice Similarity Coefficient (DSC) and 95th percentile Hausdorff distance (HD95). For the NLST dataset, we additionally report the target registration error (TRE) between landmarks. In the synthetic deformation experiments, we also report the average end-point error (AEE) to the ground truth deformation. We calculate the percentage of points with a negative Jacobian determinant $|\nabla\phi| < 0$, indicating the extent of space folding, to measure the regularity of the transformation.

### 3.1.1 EXPERIMENT 1: LARGE INTRA-SUBJECT SYNTHETIC TRANSFORMATIONS

We begin by investigating varying kinds of large synthetic deformations without any form of affine registration preprocessing. We create a dataset of intra-subject brain pairs with varying ranges of non-rigid deformations comprised of a combination of an affine and Brownian noise components. Although equivalent to real-world medical registration tasks, this experiment allows us to generate ground-truth deformations serving as a useful proof-of-concept to better evaluate recovery of large misalignments. First, a base component of fractal Brownian deformation is applied, followed by randomly uniformly sampled rotations, scaling, and translations along each dimension (see displacement field in Figure 3). We used the obtained ground truth deformation fields to quantitatively assess a method's ability to deformably recover large misalignments. The results reported in Table 1 demonstrate that our model consistently outperforms other baselines while producing the lowest amount of spatial folding. While other models struggle with large deformations, our geometric registration method is capable of fully deformably capturing the global transformation while still being able to model local deformations (see Figure 3).

### 3.1.2 EXPERIMENT 2: DEFORMABLE TRANSFORMATIONS ON PRE-ALIGNED IMAGES

In practice, it is common practice to pre-align scans in roughly the same coordinate system using affine registration, targeting the larger components of the transformation. This initial alignment enables a more accurate recovery of the smaller, deformable components in a subsequent step. To assess the capability of our method to recover deformable transformations, we conduct a comparative evaluation against several baseline methods in three distinct tasks. First, we evaluate our method on inter-subject registration of T1w-T1w images and on multi-contrast inter-subject T1w-T2w MRI brain images

Table 1: Quantitative results for intra-subject deformable registration using non-rigid synthetic deformations (multi-resolution Brownian) alongside varying degrees of uniformly-sampled rotations, scalings, and translations. Lowest setting in Brownian experiment row is used as default across all other rigid rows. Experiments consist of 100 subjects, each sampled using 10 different deformations. The performance of GeoReg with bilinear feature warping instead of a learned interpolation component $\delta$ is shown under 'feat. warp'.

| | # Param | HD95↓ | AEE$_{\phi GT}$↓ | Folding (%)↓ | HD95↓ | AEE$_{\phi GT}$↓ | Folding (%)↓ | HD95↓ | AEE$_{\phi GT}$↓ | Folding (%)↓ |
|---|---|---|---|---|---|---|---|---|---|---|
| **Brownian** | | Up to 16.41 pixels per axis (Default) | | | Up to 25.25 pixels per axis | | | Up to 33.98 pixels per axis | | |
| Affine | - | 4.695±0.979 | 1.813±0.316 | - | 4.821±0.992 | 2.638±0.452 | - | 8.188±1.561 | 2.749±0.472 | - |
| MIRTK | - | 1.940±0.170 | **1.256±0.154** | 0.000±0.000 | 1.117±0.953 | 1.981±0.232 | 0.013±0.034 | 2.336±3.243 | 2.635±0.239 | 0.113±0.148 |
| ANTs | - | 2.231±0.473 | 2.996±0.313 | 0.163±0.081 | 2.781±0.815 | 4.162±0.545 | 0.282±0.065 | 4.785±1.452 | 5.773±0.552 | 0.533±0.249 |
| LDDMM | - | 1.012±0.104 | 1.597±0.195 | 0.000±0.000 | **1.053±0.023** | 2.443±0.238 | 0.000±0.000 | **1.604±0.935** | 4.976±0.412 | 0.195±0.015 |
| VoxelMorph | 320 k | 1.656±0.159 | 3.561±0.245 | 0.003±0.002 | 3.672±0.791 | 8.323±1.282 | 0.132±0.161 | 6.199±1.910 | 11.466±1.502 | 0.249±0.203 |
| LapIRN | 924 k | −±− | −±− | −±− | −±− | −±− | −±− | −±− | −±− | −±− |
| TransMorph | 46.8 M | 1.010±0.025 | 2.247±0.049 | 1.048±0.165 | 1.085±0.088 | 3.468±0.109 | 1.998±0.212 | 1.464±0.204 | 3.960±0.079 | 3.008±0.328 |
| D-PRNet | 1.2 M | 1.081±0.097 | 2.411±0.041 | 0.856±0.166 | 1.424±0.175 | 3.306±0.100 | 1.645±0.202 | 2.552±0.476 | 3.915±0.091 | 2.883±0.247 |
| RCN | 282 M | **1.002±0.009** | 2.216±0.045 | 1.134±0.176 | 1.087±0.074 | 2.960±0.074 | 2.249±0.213 | 2.818±0.363 | 5.125±0.102 | 1.655±0.228 |
| FourierNet | 1.1 M | 1.044±0.050 | 2.444±0.069 | 0.000±0.000 | 1.350±0.123 | 3.444±0.098 | 0.001±0.001 | 1.791±0.127 | 4.669±0.137 | 0.002±0.003 |
| **Ours** (feat. warp) | 1.5 M | 2.621±0.502 | 1.637±0.161 | 0.000±0.000 | 3.923±0.594 | 2.638±0.452 | 0.000±0.000 | 3.939±1.150 | 2.801±0.280 | 0.000±0.000 |
| **Ours** (GeoReg) | 1.7 M | 1.347±0.397 | 1.328±0.152 | 0.000±0.000 | 1.763±0.421 | **1.831±0.193** | 0.000±0.000 | 2.460±0.591 | **2.580±0.303** | 0.000±0.000 |
| **Rotation + Brownian** | | ±11.25° per axis | | | ±22.5° per axis | | | ±45.0° per axis | | |
| Affine | - | 4.573±0.291 | 3.686±0.098 | - | 4.599±0.331 | 3.682±0.109 | - | 4.600±0.369 | 3.809±0.110 | - |
| MIRTK | - | **1.041±0.124** | 3.685±3.029 | 0.031±0.082 | 3.515±4.905 | 9.78±9.292 | 0.265±0.392 | 6.839±8.975 | 8.851±7.453 | 0.160±0.204 |
| ANTs | - | 3.870±1.491 | 7.011±3.616 | 0.188±0.077 | 8.813±2.152 | 18.571±3.813 | 0.215±0.088 | 11.617±5.625 | 30.327±14.763 | 0.485±0.368 |
| LDDMM | - | 1.150±0.012 | 5.765±3.619 | 0.000±0.000 | **1.041±0.124** | 13.301±6.466 | 0.000±0.000 | 5.843±6.615 | 34.077±7.573 | 0.013±0.049 |
| VoxelMorph | 320 k | 1.816±0.298 | 6.673±1.054 | 0.034±0.020 | 3.474±0.713 | 13.591±2.318 | 0.097±0.041 | 8.997±2.353 | 27.090±5.130 | 0.292±0.077 |
| LapIRN | 924 k | −±− | −±− | −±− | −±− | −±− | −±− | −±− | −±− | −±− |
| TransMorph | 46.8 M | 1.057±0.073 | 5.087±0.775 | 3.030±0.514 | 1.420±0.385 | 11.334±2.999 | 3.560±0.414 | 5.747±2.289 | 26.394±5.301 | 4.012±0.329 |
| D-PRNet | 1.2 M | 1.557±0.367 | 7.002±1.202 | 1.422±0.188 | 3.580±1.082 | 14.058±2.896 | 1.629±0.265 | 9.200±2.444 | 28.278±5.769 | 2.192±0.313 |
| RCN | 282 M | 1.364±0.130 | 4.262±0.518 | 3.640±0.698 | 1.902±0.218 | 11.082±2.042 | 3.945±0.558 | 4.951±1.777 | 26.537±6.120 | 4.029±0.505 |
| FourierNet | 1.1 M | 2.224±1.205 | 8.804±4.194 | 0.000±0.000 | 4.875±3.666 | 16.706±8.034 | 0.000±0.000 | 14.253±5.661 | 36.493±13.914 | 0.007±0.016 |
| **Ours** (feat. warp) | 1.5 M | 2.068±0.484 | 1.585±0.312 | 0.000±0.000 | 2.620±1.358 | 1.989±0.753 | 0.000±0.000 | 2.818±0.546 | 2.477±0.928 | 0.000±0.000 |
| **Ours** (GeoReg) | 1.7 M | 1.520±0.332 | **1.511±0.260** | 0.000±0.000 | 1.630±0.415 | **1.604±0.363** | 0.000±0.000 | 2.054±0.385 | **1.951±0.598** | 0.026±0.145 |
| **Scaling + Brownian** | | ±10% of image size per axis | | | ±30% of image size per axis | | | ±50% of image size per axis | | |
| Affine | - | 4.529±0.368 | 3.685±0.101 | - | 4.891±0.431 | 3.687±0.113 | - | 5.138±0.515 | 3.749±0.116 | - |
| MIRTK | - | 1.052±0.127 | 1.462±0.348 | 0.039±0.002 | 4.545±1.598 | 1.554±0.284 | 0.398±0.699 | 9.780±11.523 | 13.425±11.322 | 0.581±0.838 |
| ANTs | - | 3.124±0.584 | 0.584±1.113 | 0.202±0.117 | 9.343±4.464 | 14.023±4.234 | 0.242±0.105 | 14.641±4.983 | 20.269±4.888 | 0.191±0.097 |
| LDDMM | - | 1.563±0.342 | 2.497±0.700 | 0.000±0.000 | 1.902±0.235 | 4.765±2.273 | 0.000±0.000 | 1.883±0.166 | 8.979±4.21 | 0.000±0.000 |
| VoxelMorph | 320 k | 1.706±0.167 | 3.542±0.242 | 0.002±0.002 | 3.389±0.642 | 7.074±1.045 | 0.122±0.130 | 7.592±2.113 | 12.287±1.798 | 0.228±0.116 |
| LapIRN | 924 k | −±− | −±− | −±− | −±− | −±− | −±− | −±− | −±− | −±− |
| Transmorph | 46.8 M | 1.074±0.079 | 3.308±0.214 | 1.536±0.213 | 1.370±0.363 | 6.843±0.635 | 3.753±0.612 | 3.154±1.531 | 10.307±1.705 | 5.086±0.621 |
| D-PRNet | 1.2 M | 1.250±0.137 | 3.598±0.270 | 1.388±0.318 | 2.225±0.347 | 7.505±1.028 | 3.200±0.445 | 5.391±2.160 | 11.986±2.402 | 3.910±0.465 |
| RCN | 282 M | 1.337±0.188 | 3.307±0.181 | 1.600±0.370 | 2.593±0.252 | 5.396±0.586 | 4.642±0.806 | 3.785±0.661 | 7.834±1.276 | 5.644±0.687 |
| FourierNet | 1.1 M | 1.307±0.302 | 3.831±0.601 | 0.000±0.000 | 5.068±4.076 | 9.190±3.395 | 0.062±0.069 | 10.102±4.565 | 20.638±4.237 | 0.114±0.123 |
| **Ours** (feat. warp) | 1.5 M | 1.910±0.355 | 1.490±0.277 | 0.000±0.000 | 2.566±0.617 | 2.073±0.925 | 0.000±0.000 | 2.961±0.796 | 2.400±1.290 | 0.000±0.000 |
| **Ours** (GeoReg) | 1.7 M | **1.040±0.122** | 1.375±0.357 | 0.000±0.000 | **1.274±0.312** | 1.714±0.904 | 0.000±0.000 | 1.486±0.523 | 2.234±1.586 | 0.000±0.000 |
| **Translation + Brownian** | | ±10% of image size per axis | | | ±30% of image size per axis | | | ±50% of image size per axis | | |
| Affine | - | 4.791±1.106 | 2.092±0.362 | - | 4.683±1.241 | 2.076±0.386 | - | 4.768±1.088 | 2.025±0.497 | - |
| MIRTK | - | 2.217±0.256 | 1.583±0.418 | 0.030±0.172 | 15.738±9.906 | 15.912±7.513 | 0.920±0.549 | 31.954±18.117 | 32.458±18.79 | 0.557±0.413 |
| ANTs | - | 2.641±1.983 | 4.289±1.888 | 0.191±0.097 | 20.516±6.970 | 21.84±7.159 | 0.206±0.139 | 39.502±13.644 | 42.077±14.147 | 0.139±0.053 |
| LDDMM | - | 1.962±0.310 | 3.195±0.806 | 0.000±0.000 | 15.476±10.564 | 14.518±6.879 | 0.000±0.000 | 31.702±17.558 | 30.984±13.721 | 0.549±1.403 |
| VoxelMorph | 320 k | 3.468±0.529 | 5.436±0.495 | 0.033±0.034 | 18.075±2.677 | 16.027±1.977 | 0.628±0.194 | 31.645±4.145 | 26.826±3.932 | 1.479±0.395 |
| LapIRN | 924 k | −±− | −±− | −±− | −±− | −±− | −±− | −±− | −±− | −±− |
| TransMorph | 46.8 M | 1.641±0.385 | 6.065±0.634 | 2.601±0.421 | 17.865±4.339 | 20.212±3.201 | 4.378±0.275 | 40.148±7.138 | 37.221±5.569 | 5.920±0.147 |
| D-PRNet | 1.2 M | 3.720±0.879 | 6.045±0.751 | 2.631±0.483 | 5.477±0.692 | 12.834±1.757 | 5.556±0.428 | 6.669±0.833 | 17.522±2.954 | 6.636±0.509 |
| RCN | 282 M | 2.217±0.193 | 3.559±0.163 | 4.465±1.071 | 4.632±0.352 | 6.427±0.608 | 6.491±0.883 | 5.123±0.430 | 10.281±1.381 | 7.166±0.565 |
| FourierNet | 1.1 M | 1.790±0.139 | 2.920±0.062 | 0.019±0.029 | 4.762±0.325 | 3.737±0.116 | 0.146±0.109 | 4.664±0.470 | 4.032±0.177 | 0.087±0.074 |
| **Ours** (feat. warp) | 1.5 M | 2.193±0.334 | 1.474±0.156 | 0.000±0.000 | 3.397±0.446 | 1.949±0.189 | 0.000±0.000 | 4.605±2.827 | 3.279±2.685 | 0.000±0.000 |
| **Ours** (GeoReg) | 1.7 M | **1.293±0.308** | **1.288±0.161** | 0.000±0.000 | **1.603±0.329** | **1.434±0.205** | 0.000±0.000 | 2.260±0.358 | **1.760±0.295** | 0.000±0.000 |
| **Affine + Brownian** | | ±11.25° Rot., ±10% Scale, ±10% Transl. | | | ±22.5° Rot., ±30% Scale, ±30% Transl. | | | ±45.0° Rot., ±50% Scale, ±50% Transl. | | |
| Affine | - | 4.646±1.339 | 3.106±1.009 | - | 4.741±1.381 | 4.612±0.294 | - | 4.931±2.165 | 6.388±3.766 | - |
| MIRTK | - | 1.182±0.548 | 4.348±2.152 | 0.039±0.102 | 14.953±13.197 | 17.389±10.896 | 0.442±0.456 | 54.075±13.242 | 45.730±13.964 | 1.188±0.882 |
| ANTs | - | 8.933±2.450 | 11.500±3.646 | 0.170±0.071 | 22.646±6.038 | 27.093±5.549 | 0.159±0.059 | 45.812±13.148 | 53.287±7.424 | 0.523±0.535 |
| LDDMM | - | 1.885±2.655 | 8.062±3.118 | 0.000±0.000 | 16.810±10.331 | 21.628±7.729 | 0.064±0.193 | 39.410±16.364 | 51.512±12.734 | 0.061±0.139 |
| VoxelMorph | 320 k | 3.144±2.181 | 4.464±1.065 | 0.034±0.061 | 23.367±12.870 | 9.264±1.842 | 0.983±0.714 | 34.688±15.383 | 13.938±5.523 | 2.237±1.810 |
| LapIRN | 924 k | −±− | −±− | −±− | −±− | −±− | −±− | −±− | −±− | −±− |
| TransMorph | 46.8 M | 1.215±0.459 | 6.221±1.813 | 4.738±0.926 | 3.489±3.174 | 15.329±6.015 | 20.046±16.595 | 39.186±15.634 | 30.938±11.596 | 7.950±1.028 |
| D-PRNet | 1.2 M | 4.081±1.298 | 9.402±0.939 | 0.258±0.262 | 11.621±2.591 | 19.445±3.051 | 6.128±0.506 | 27.491±5.682 | 41.544±5.765 | 6.709±0.318 |
| RCN | 282 M | **1.023±0.034** | 3.749±0.297 | 5.368±0.355 | **2.006±0.160** | 9.279±0.932 | 7.120±0.305 | 5.014±1.692 | 23.953±4.785 | 7.563±0.510 |
| FourierNet | 1.1 M | 1.469±0.096 | 2.825±0.070 | 0.001±0.001 | 2.496±0.203 | 3.561±0.115 | 0.003±0.005 | 4.221±0.234 | **4.005±0.100** | 0.004±0.006 |
| **Ours** (feat. warp) | 1.5 M | 2.384±0.491 | 1.854±0.369 | 0.000±0.000 | 2.982±1.089 | 3.031±1.016 | 0.000±0.000 | **4.109±2.371** | 6.436±3.811 | 0.001±0.001 |
| **Ours** (GeoReg) | 1.7 M | 1.880±0.431 | **1.614±0.239** | 0.000±0.000 | 3.567±0.960 | **2.576±0.703** | 0.000±0.000 | 6.275±3.026 | 4.437±2.774 | 0.063±0.347 |

from the CamCAN dataset. Next, we recover intra-subject inhale-exhale motion using the National Lung Screening Trial (NLST) dataset. Results outlined in Table 2 show our model demonstrates accuracy on par with the state-of-the-art across all registration tasks using fewer learnable parameters.

## 3.2 DISCUSSION AND FUTURE WORK

In the absence of ground truth deformation fields, registration algorithms are evaluated using surrogate measures of accuracy that compete against transformation regularity. As a result, one has to resort to inspecting the registration results qualitatively. This can be observed in Table 1 in the rotation experiments, where Transmorph demonstrates high HD95 performance while suffering from a high folding ratio. The inadequacy of the transformation becomes prominent in Figure 3, where Transmorph displays a poor qualitative registration result. On the other hand, we believe our approaches' top-down composition of the transformation by predicting hierarchical refinements provides a strong architectural prior to regularizing a registration task. This also explains our method's high perfor-

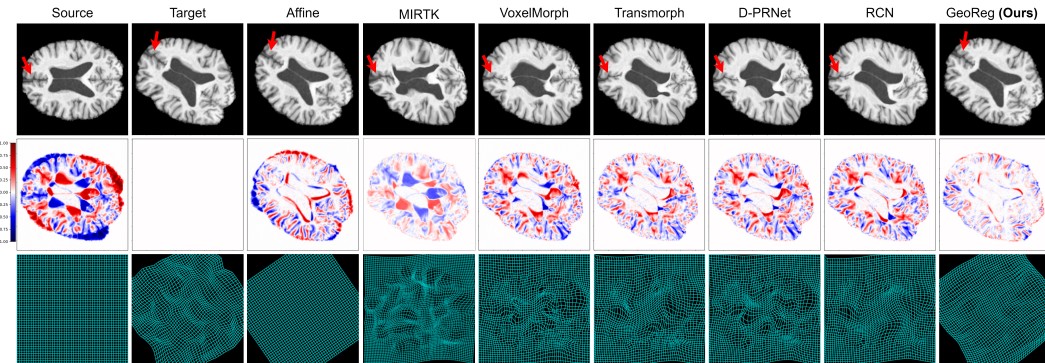

Figure 3: Qualitative results for an intra-subject $(45°, 0°, 0°)$ rotation experiment with added random Brownian noise deformation. Red arrows indicate the same structure across all methods. Our method is able to recover affine and deformable components despite modeling the transformation fully deformably.

Table 2: Quantitative results measuring the accuracy and regularity of different registration methods on brain T1w inter-subject (CamCAN) and inhale-exhale lung CT (NLST) registration. The performance of GeoReg with bilinear feature warping instead of a learned interpolation component $\delta$ is shown under 'feat. warp'. The asterisk * denotes non statistically significant results of GeoReg over a given baseline ($p > 0.05$).

| | | Brain CamCAN | | | Brain CamCAN T1T2 | | | Lung NLST | | | |
|---|---|---|---|---|---|---|---|---|---|---|---|
| | # Param. | DSC ↑ | HD95 ↓ | Folding (%) ↓ | DSC ↑ | HD95 ↓ | Folding (%) ↓ | DSC ↑ | HD95 ↓ | TRE↓ | Folding (%) ↓ |
| Affine | - | $0.604 \pm 0.04$ | $4.12 \pm 0.07$ | - | $0.604 \pm 0.04$ | $4.12 \pm 0.07$ | - | $0.928 \pm 0.09$ | $4.24 \pm 0.05$ | $9.07 \pm 0.12$ | - |
| MIRTK [FFD] | - | $0.836 \pm 0.03^*$ | $2.65 \pm 0.48$ | $0.10 \pm 0.08$ | $0.721 \pm 0.06$ | $3.65 \pm 1.22$ | $0.05 \pm 0.03$ | $0.971 \pm 0.02^*$ | $1.31 \pm 0.68$ | $\mathbf{2.76 \pm 0.05}$ | $0.03 \pm 0.01$ |
| ANTs [SyN] | - | $0.808 \pm 0.06$ | $2.55 \pm 0.08^*$ | $0.15 \pm 0.05$ | $0.767 \pm 0.02^*$ | $3.05 \pm 0.90^*$ | $0.10 \pm 0.01$ | $0.973 \pm 0.02^*$ | $1.09 \pm 0.12^*$ | $3.15 \pm 1.05$ | $0.02 \pm 0.01$ |
| LDDMM | - | $0.820 \pm 0.06$ | $2.85 \pm 0.88^*$ | $0.12 \pm 0.01$ | $0.752 \pm 0.06$ | $3.21 \pm 0.95^*$ | $0.08 \pm 0.01$ | $\mathbf{0.984 \pm 0.02}^*$ | $1.25 \pm 0.97^*$ | $3.06 \pm 1.15$ | $0.02 \pm 0.03$ |
| VoxelMorph | 320 k | $0.806 \pm 0.02$ | $3.60 \pm 0.93$ | $0.32 \pm 0.05$ | $0.753 \pm 0.03$ | $3.88 \pm 2.01$ | $0.23 \pm 0.03$ | $0.971 \pm 0.02^*$ | $2.01 \pm 1.30$ | $5.46 \pm 0.66$ | $0.13 \pm 0.10$ |
| LapIRN | 924 k | $0.820 \pm 0.04$ | $3.06 \pm 1.40$ | $0.31 \pm 0.02$ | $0.758 \pm 0.04$ | $3.57 \pm 2.27$ | $0.33 \pm 0.02$ | $0.976 \pm 0.02$ | $1.12 \pm 0.56$ | $3.12 \pm 1.01$ | $0.12 \pm 0.05$ |
| Transmorph | 46.8 M | $0.826 \pm 0.01$ | $2.75 \pm 0.81$ | $0.45 \pm 0.02$ | $0.768 \pm 0.04^*$ | $3.04 \pm 0.23^*$ | $0.32 \pm 0.06$ | $0.975 \pm 0.04^*$ | $1.45 \pm 0.53$ | $4.59 \pm 0.60$ | $0.20 \pm 0.16$ |
| D-PRNet | 1.2 M | $0.828 \pm 0.02$ | $2.89 \pm 0.94$ | $0.47 \pm 0.04$ | $0.733 \pm 0.04$ | $4.63 \pm 1.98$ | $0.28 \pm 0.03$ | $0.970 \pm 0.05$ | $1.92 \pm 1.00$ | $4.76 \pm 0.62$ | $0.19 \pm 0.12$ |
| RCN | 282 M | $0.814 \pm 0.01$ | $2.63 \pm 0.58^*$ | $0.17 \pm 0.02$ | $0.763 \pm 0.28^*$ | $3.21 \pm 1.65^*$ | $0.25 \pm 0.05$ | $0.965 \pm 0.08^*$ | $2.56 \pm 1.54$ | $4.60 \pm 0.64$ | $0.11 \pm 0.09$ |
| FourierNet | 1.1 M | $0.821 \pm 0.06$ | $2.52 \pm 0.57^*$ | $0.18 \pm 0.01$ | $0.767 \pm 0.18^*$ | $3.01 \pm 1.60^*$ | $0.25 \pm 0.05$ | $0.973 \pm 0.02$ | $1.14 \pm 0.12$ | $3.15 \pm 1.24$ | $0.22 \pm 0.01$ |
| **Ours** (feat. warp) | 657 k | $0.812 \pm 0.07$ | $2.51 \pm 0.95^*$ | $0.13 \pm 0.09$ | $0.727 \pm 0.07$ | $\mathbf{2.95 \pm 1.16}^*$ | $0.27 \pm 0.03$ | $0.962 \pm 0.09^*$ | $1.38 \pm 0.41$ | $5.32 \pm 2.20$ | $0.32 \pm 0.02$ |
| **Ours** (GeoReg) | 741 k | $\mathbf{0.838 \pm 0.06}$ | $\mathbf{2.45 \pm 0.82}$ | $0.42 \pm 0.06$ | $\mathbf{0.778 \pm 0.09}$ | $2.98 \pm 0.89$ | $0.16 \pm 0.04$ | $0.972 \pm 0.03$ | $\mathbf{1.08 \pm 0.16}$ | $3.62 \pm 0.63$ | $0.23 \pm 0.09$ |

mance despite its comparatively low number of learnable parameters. Despite optimizations on many aspects of the graph represented in memory, our method has a substantially larger memory footprint than its grid-based counterparts due to having to store explicit intermediate coordinates. Nonetheless, our layers' ability to directly incorporate volume spacing into the deformation prediction could present an interesting avenue to overcome the limitations of current registration approaches in anisotropic tasks. Finally, this work has only explored large deformations under synthetically transformed intra-subject brain data. However, these synthetic deformations might not capture all the intricate differences of inter-subject registration scenarios. We plan to explore this in future work.

## 4 CONCLUSION

In this work, we introduce a novel formulation of deformable image registration by using geometric deep-learning principles. We discuss the benefits of estimating deformations on non-fixed grid locations by defining data-driven functions on continuous domains. We outline the need for two types of learned graph operations: A deformation modeling function $\tau$ and a cross-resolution interpolation function $\delta$. Our multi-resolution architecture shows the ability to fully deformably capture a wide range of rotations, translations, and scalings without explicitly modeling an affine component.

We believe that even though our work provides competitive results with state-of-the-art methods, the main contribution of our manuscript lies in establishing a theoretical foundation by which transformations can be explicitly propagated through a deep learning architecture. We think that this contribution opens up avenues of research to reduce the black-box nature of current learned registration paradigms, incorporate ideas from conventional image registration into deep learning architectures, and tackle known data issues such as anisotropicity.

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
