APPENDIX

# A ARCHITECTURAL OVERVIEW

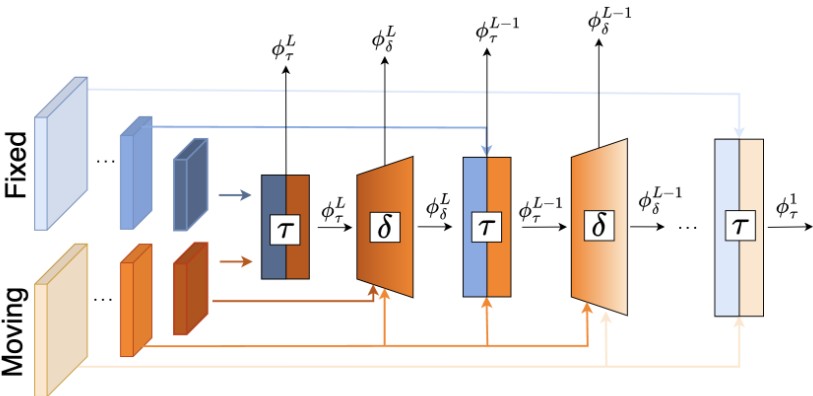

Figure 4: Architectural overview of the proposed method.

# B PSEUDOCODE OF FUNCTIONS $\tau$ AND $\delta$

---

**Algorithm 1:** Pseudocode of implementation of deformation function $\tau$ for a given decoder resolution

---

Function $\tau$ $(\boldsymbol{F}^S, \boldsymbol{X}^S, \boldsymbol{F}^T, \boldsymbol{X}^T, \theta^\tau)$:

**Input:** Features and coordinates of source nodes $S$ and target nodes $T$. Learnable parameters $\theta^\tau$.
**Output:** Deformations $\boldsymbol{\Phi}$ for source nodes $S$

// Initialize deformations as zeros
$\boldsymbol{\Phi} \leftarrow \boldsymbol{0}^{S \times 3}$
// Repeat deformation N times
$n \leftarrow 0$
**while** $n < N$ **do**
  // Perform node-wise deformation on source domain
  **for** $s \in S$ **do**
    $\mathbf{f}_s \leftarrow \boldsymbol{F}^S[s]$
    $\mathbf{x}_s \leftarrow \boldsymbol{X}^S[s] + \boldsymbol{\Phi}[s]$
    // Define neighborhood for current source node $s$
    $\mathcal{N}_s^T \leftarrow \text{KNN}(\mathbf{x}_s, \boldsymbol{X}^T)$
    // Perform cross-attention between a source node and its neighborhood
    $\boldsymbol{F}^{\mathcal{N}_s^T} \leftarrow \boldsymbol{F}^T[\mathcal{N}_s^T] + \text{POSENCODE}(\boldsymbol{X}^T[\mathcal{N}_s^T] - \mathbf{x}_s)$
    $\mathbf{f}_s' \leftarrow \text{CROSSATTENTION}(\mathbf{f}_s, \boldsymbol{F}^{\mathcal{N}_s^T}; \theta^\tau)$
    // Map output features into deformation vector and update total deformation
    $\phi \leftarrow \text{LINEAR}(\mathbf{f}_s'; \theta^\tau)$
    // Update total transformation estimate
    $\boldsymbol{\Phi}[s] \leftarrow \boldsymbol{\Phi}[s] + \phi$
  **end for**
  $n \leftarrow n + 1$
**end while**
return $\boldsymbol{\Phi}$

---

---

**Algorithm 2:** Pseudocode of implementation of deformation interpolation function $\delta$ between decoder resolution layers $l$ and $l-1$

---

$\underline{\text{Function } \delta}\ (\boldsymbol{F}^l, \boldsymbol{X}^l, \boldsymbol{\Phi}^l, \boldsymbol{F}^{l-1}, \boldsymbol{X}^{l-1}, \theta^\delta)$:

**Input:** Features $\boldsymbol{F}^l$, starting coordinates $\boldsymbol{X}^l$, and of parent source nodes $S$ in layer $l$. Deformations $\boldsymbol{\Phi}^l$ for parent nodes $S$ in layer $l$. Features $\boldsymbol{F}^{l-1}$ and coordinates $\boldsymbol{X}^{l-1}$ of child source nodes $S$ in layer $l-1$. Learnable parameters $\theta^\delta$.
**Output:** Deformations $\boldsymbol{\Phi}^{l-1}$ for nodes in layer $l-1$

    // Initialize deformations as zeros
    $\boldsymbol{\Phi}^{l-1} \leftarrow \mathbf{0}^{S^{l-1} \times 3}$
    // Perform node-wise deformation interpolation on layer $l-1$
    **for** $s \in S^{l-1}$ **do**
        $\mathbf{f} \leftarrow \boldsymbol{F}^{l-1}[s]$
        $\mathbf{x} \leftarrow \boldsymbol{X}^{l-1}[s]$
        // Define neighborhood of child node $s$ prior to any deformations to parent nodes $S^l$.
        $\mathcal{N}_s^{S^l} \leftarrow \text{KNN}(\mathbf{x}, \boldsymbol{X}^l)$
        // Compute initial deformation estimate using neighborhood mean
        $\phi_{inh} \leftarrow \text{MEAN}(\boldsymbol{\Phi}^l[\mathcal{N}_s^{S^l}])$
        $\mathbf{x}' \leftarrow \mathbf{x} + \phi_{inh}$
        // Perform cross-attention between child node $s$ and its neighbourhood of parent nodes $\mathcal{N}_s^{S^l}$
        $\boldsymbol{X}^{\mathcal{N}_s^{S^l}} \leftarrow \boldsymbol{X}^l[\mathcal{N}_s^{S^l}] + \boldsymbol{\Phi}^l[\mathcal{N}_s^{S^l}]$
        $\boldsymbol{F}^{\mathcal{N}_s^{S^l}} \leftarrow \boldsymbol{F}^l[\mathcal{N}_s^{S^l}] + \text{POSENCODE}(\boldsymbol{X}^{\mathcal{N}_s^{S^l}} - \mathbf{x}')$
        $\mathbf{f}' \leftarrow \text{CROSSATTENTION}(\mathbf{f}, \boldsymbol{F}^{\mathcal{N}_s^{S^l}}; \theta^\delta)$
        // Map output features into deformation vector
        $\phi_{interp} \leftarrow \text{LINEAR}(\mathbf{f}'; \theta^\delta)$
        $\boldsymbol{\Phi}^{l-1}[s] \leftarrow \phi_{inh} + \phi_{interp}$
    **end for**
    return $\boldsymbol{\Phi}^{l-1}$

---

## C    DATA PRE-PROCESSING

Dataset, pre-processing and label information of the CamCAN (Shafto et al., 2014; Taylor et al., 2017) and the NLST (team, 2011) datasets. CamCAN dataset consists of 310 T1w and T2w MR 3D images ($160 \times 180 \times 160$, $1\text{mm}^3$ isotropic resolution). Preprocessing includes normalization to MNI (Horn, 2016) space using affine registration, skull-stripping with ROBEX (Iglesias et al., 2011), and bias-field correction with SimpleITK (Lowekamp et al., 2013). Automated segmentation of 138 cortical and subcortical structures (categorized into five groups for reporting) was performed using MALPEM (Ledig et al., 2015). NLST consists of 150 pairs of CT scans ($224 \times 192 \times 224$, $1.5\,\text{mm}$ isotropic) at inspiration and expiration phases. The lungs were automatically segmented, and landmarks were automatically extracted. We refer to the Learn2Reg Challenge 2022 for the pre-processing, lung segmentation, and keypoint extraction details. For training, validation, and testing, we use $80\% - 10\% - 10\%$ splits for both datasets.

# D TRAINING MEMORY FOOTPRINTS

Table 3: VRAM requirements per model in GigaBytes (GB) under a batch size of 1.

| Models | VRAM |
|---|---|
| VoxelMorph | 3.55 GB |
| LapIRN | 6.67 GB |
| Transmorph | 7.09 GB |
| D-PRNet | 11.11 GB |
| RCN | 6.21 GB |
| FourierNet | 1.53 GB |
| **Ours** (feat. warp) | 6.75 GB |
| **Ours** (GeoReg) | 9.08 GB |

# E QUALITATIVE RESULTS ON SYNTHETIC AFFINE TRANSFORMATIONS

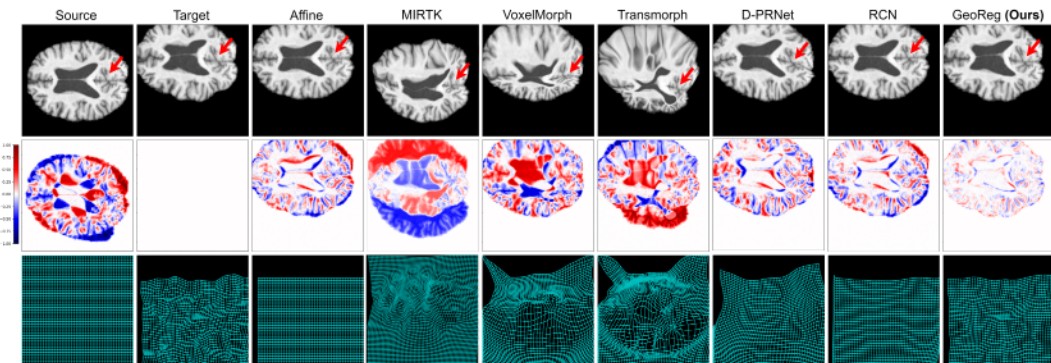

Figure 5: Qualitative results of all compared methods for an inter-subject $(25\%, 5\%, 0\%)$ of image shape translation registration experiment with added random Brownian noise deformation. Red arrows indicate the same brain structure across all registration methods. Our method is able to recover affine and deformable components despite modeling the transformation fully deformably.

## F  LAYER-WISE VISUALIZATION OF TRANSFORMATION COMPONENTS

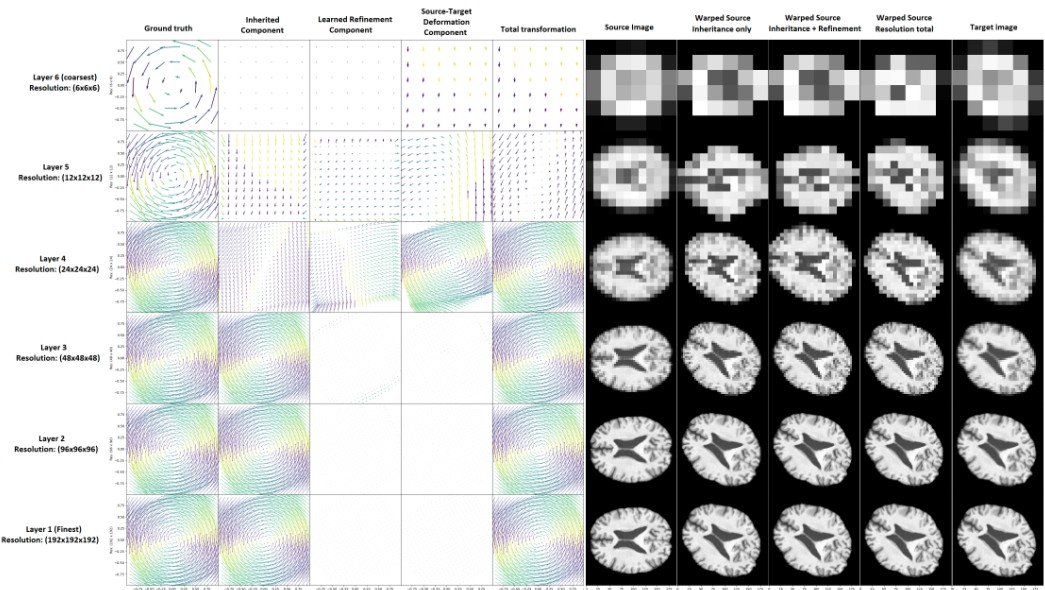

Figure 6: Per-layer deformations components as predicted by the coarse-to-fine decoding process (top-down in the figure's rows). Coarser layers manage to model the majority of the transformation, resulting in finer layers not having to produce meaningful deformations as their inherited transformations are already very close to ground truth transformation.

## G  QUALITATIVE RESULTS OF THE DEFORMABLE REGISTRATION

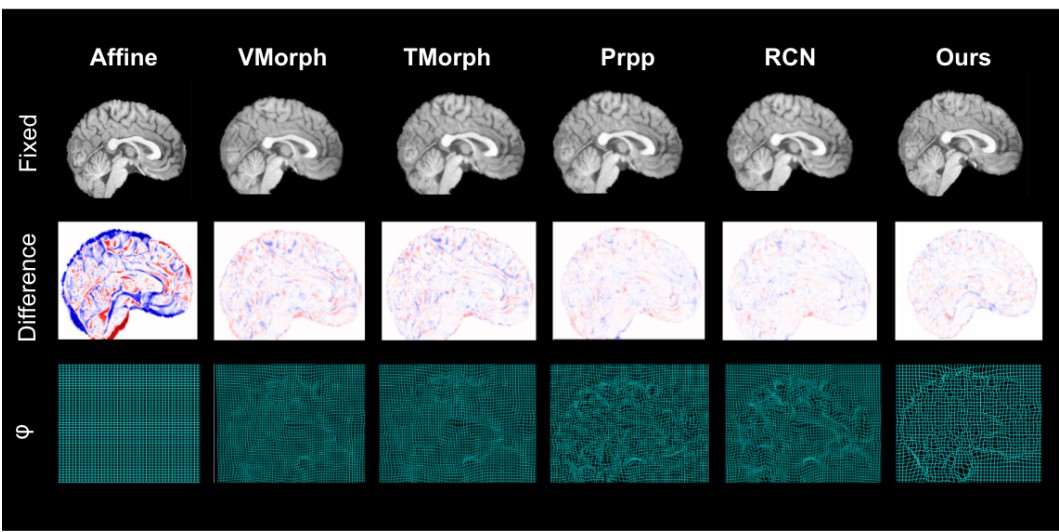

Figure 7: Qualitative results of all compared methods for the CamCAN T1w-T1w inter-subject deformable registration experiment.

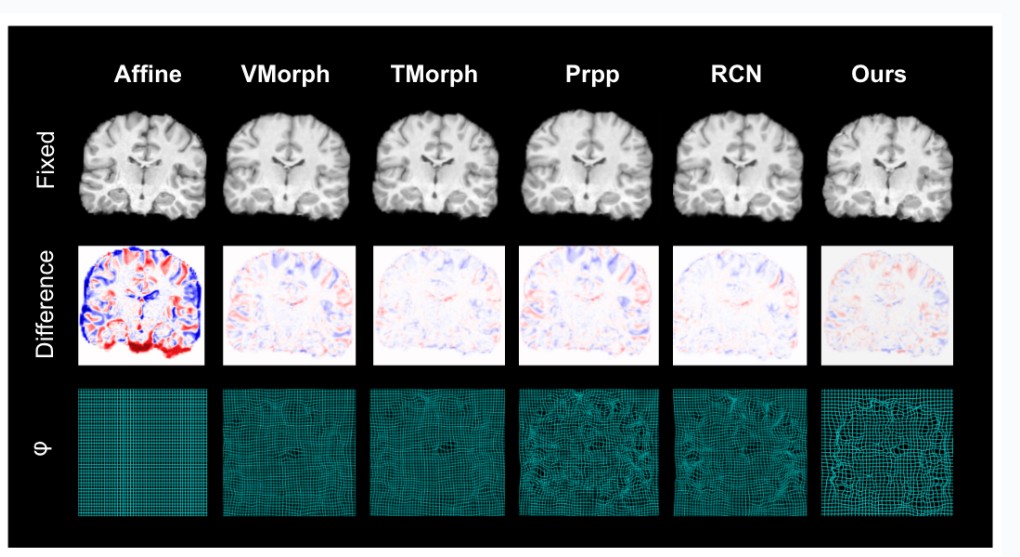

Figure 8: Qualitative results of all compared methods for the CamCAN T1w-T1w inter-subject deformable registration experiment.

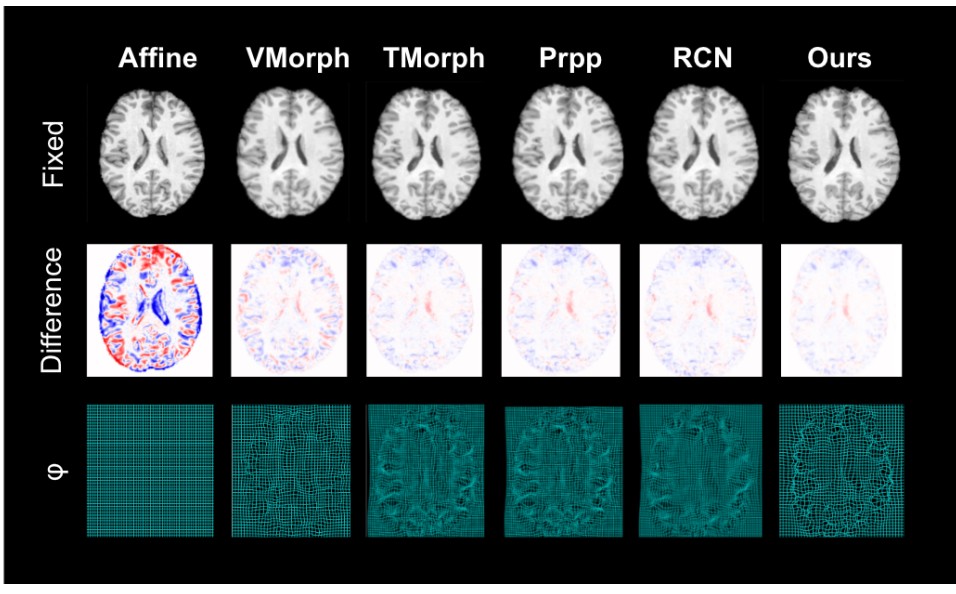

Figure 9: Qualitative results of all compared methods for the CamCAN T1w-T1w inter-subject deformable registration experiment.

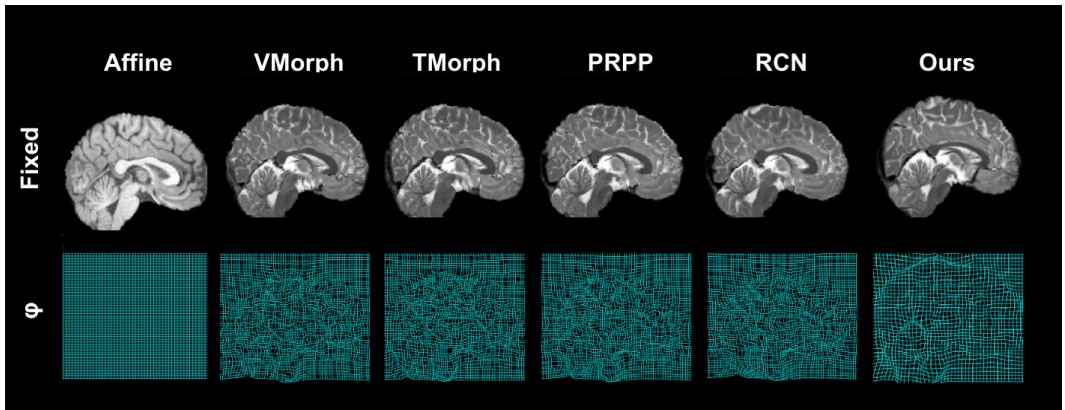

Figure 10: Qualitative results of all compared methods for the CamCAN T1w-T2w inter-subject deformable registration experiment.

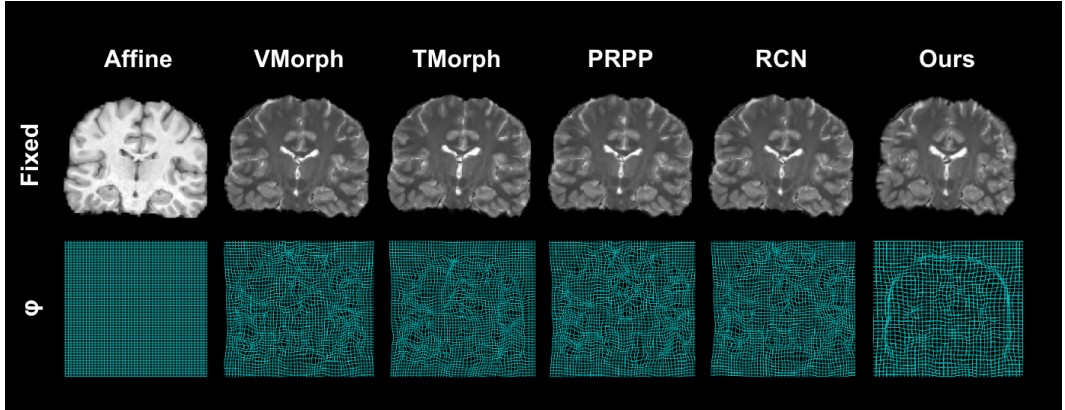

Figure 11: Qualitative results of all compared methods for the CamCAN T1w-T2w inter-subject deformable registration experiment.

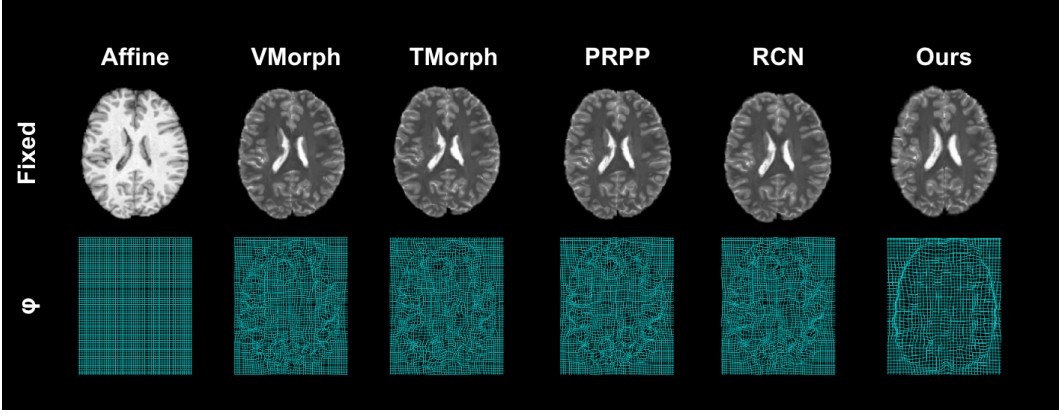

Figure 12: Qualitative results of all compared methods for the CamCAN T1w-T2w inter-subject deformable registration experiment.

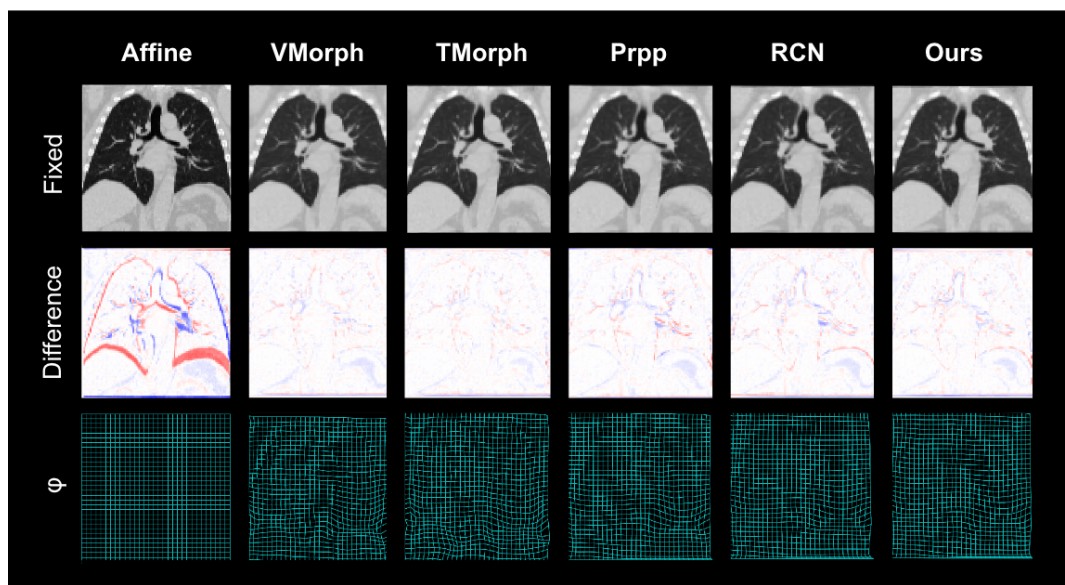

Figure 13: Qualitative results of all compared methods for the NLST inhale-exhale deformable registration experiment.

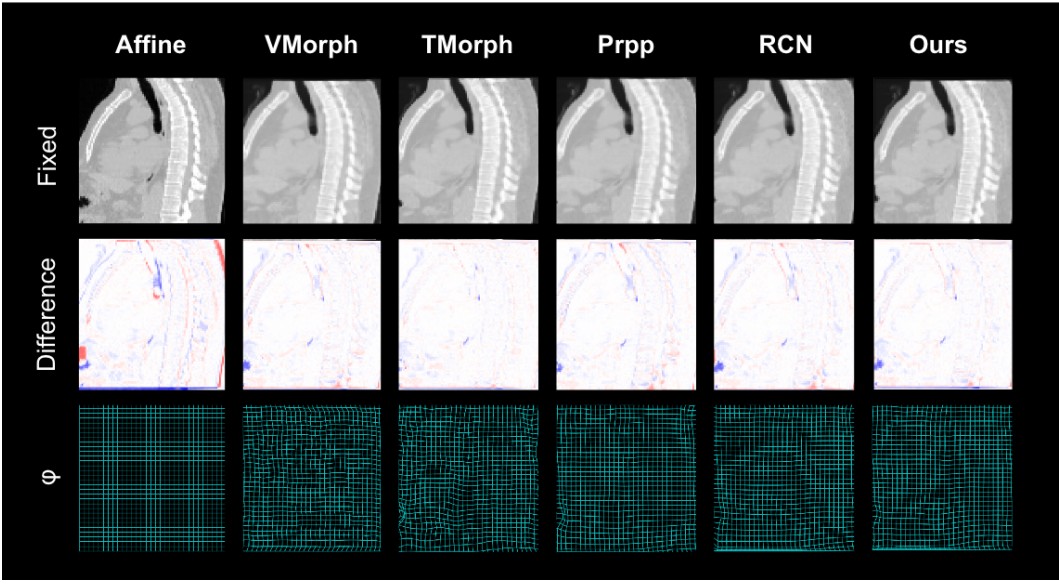

Figure 14: Qualitative results of all compared methods for the NLST inhale-exhale deformable registration experiment.

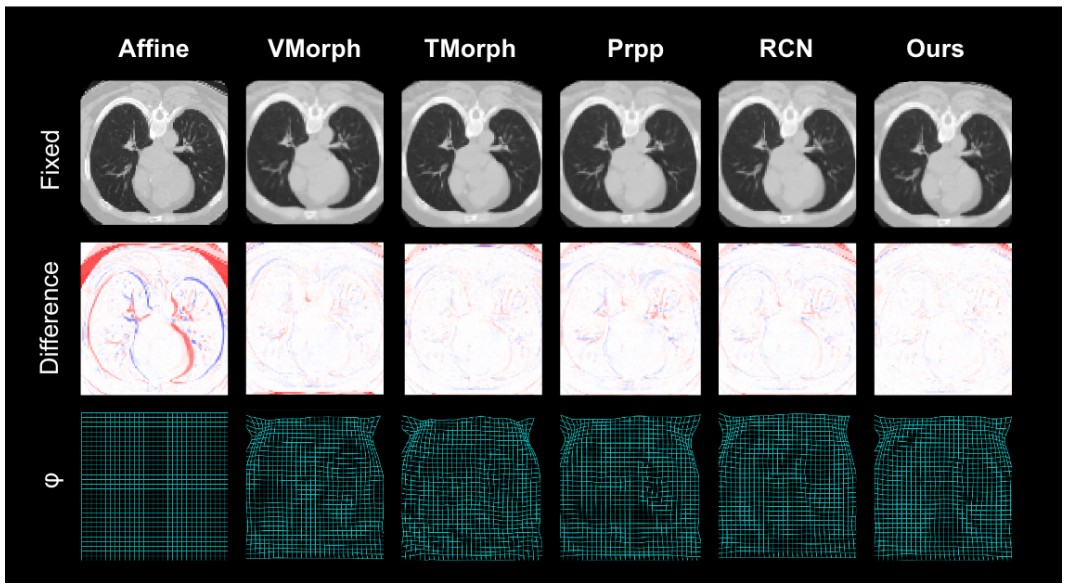

Figure 15: Qualitative results of all compared methods for the NLST inhale-exhale deformable registration experiment.