# OpenReview forum: "Image registration is a geometric deep learning task"
_ICLR.cc/2025/Conference — Submitted to ICLR 2025_

### Official Review · Reviewer_AqRJ · 2024-10-20

**Soundness:** 2
**Presentation:** 2
**Contribution:** 3
**Rating:** 5
**Confidence:** 4

**Summary:**

This paper proposes a method, GeoReg, that uses geometric deep-learning principles to model deformations without grid requirement, avoiding the need for interpolation in the feature space. Inspired by Lagrangian reference frames of the deformation field, the image features are tracked individually along the trajectory rather than observing the deformation at specific locations, and their coordinates are updated under graph operations. To overcome the local minima, the multi-resolution strategy is applied.

**Strengths:**

The first introduction of geometric deep learning and the Lagrangian reference into the learning-based method make the paper a novel and significant algorithm contribution.

The paper is generally easy to understand. Some descriptions that need to be clarified are listed later.

The experiments demonstrate the superior performance of the proposed method in the cases with rotation, scaling, and translation deformation.

**Weaknesses:**

1. The paper tries to demonstrate the proposed model’s ability to recover large deformations. Certainly, the model has superior performance in handling rotation, scaling, and translation deformation separately. In practice, the large deformation in medical images should be the combination of the three transformations, which fails to be demonstrated in the experiment. Moreover, the organs are not a rigid body, thus the deformation might not be uniform even in the same organ. For instance, the scaling of the lung at one end might be different from the other end.
2. The benchmarks used in the paper are not refined for the image registration with large deformation. The model that is specifically designed for large deformation such as LapIRN (Mok & Chung, 2020) which is mentioned in the paper, FourierNet (Jia et al, 2023), and more need to be compared to assess the performance of the proposed model on large deformation registration.
3. The experiments are all based on datasets with only one main organ (brain & lung), which might lack the deformation discontinuities of sliding organs (one type of large deformation, see Papiez et al., 2014). The datasets with multiple organs Abdomen CT-CT dataset (also available in learn2reg) might be needed to examine the proposed model's performance on various large deformations.

Minor problems
1. Table 2: In row 3, typo of the model name (VoxelMorph?)
2. Table 2: In the middle column (Brain CamCAN T1T2), should the bold font be put on the second to the last row (2.95+/-1.16) rather than the final row (2.98+/-0.89)

**Questions:**

Several suggestions that might make the paper more solid:
1. It would be better to add some models that specifically designed for large deformations
2. The first experiment could add the tests on the combination of transformations.
3. Add experiment on additional dataset with multiple organs which could contain slippery deformation.

---

> ### Author Response · Authors · 2024-11-22
>
> We would like to thank the reviewer for the encouraging remarks and feedback on strengthening the results section.
>
> > Weakness 2: Additional baselines
>
> We agree with the suggestions offered by the reviewer regarding the inclusion of LapIRN and FourierNet as baselines relevant to large deformations. We have thus extended our results section to include these baselines. Additionally, we have also added LDDMM as an additional conventional registration baseline.
>
> > Weakness 1: Full rigid transformations and local deformations
>
> Regarding the comment to extend Experiment 1 with all three transformations. Our new draft currently extends Table 1 with a full rigid transformation setting including all 3 transformations: rotation, scaling, and translation. However, we would like to point out that across all settings of Experiment 1, we apply Brownian noise deformation at multiple scales to ensure the synthetic transformation is not strictly rigid. The amount this local deformation impacts the tissue structures of Experiment 1 can be seen in the qualitative results of Figure 3 of the manuscript. We would also like to inform the reviewer, that following other comments in the review process, we also have plans to expand Experiment 1 with an additional setting where various increasing Brownian noise deformation levels are examined WITHOUT a global rigid component. (Please note that as of the posting of this comment, the current state of our draft provides partial results for some of the baselines. We plan to finish filling in these results by the end of the rebuttal period)
>
> Nonetheless, we acknowledge the aforementioned experiments are not examples of real-world tasks. They are only meant as a proof-of-concept where the ground-truth deformation is known. We have expanded the wording of Experiment 1 to more clearly reflect this fact. For a real-world example of large deformations, we would like to refer the reviewer to the inhale-exhale lung CT experiments under Experiment 2, whose qualitative result visualizations may be found in Appendix G.
>
> > Weakness 3: Abdomen CT-CT datasets
>
> Due to the challenging nature of Abdominal datasets and the compute-heavy process of finding working hyper-parameters for all baselines, we are unfortunately not able to run experiments on the suggested Abdomen CT-CT dataset within the current time frame. We would however like to acknowledge that we find Abdominal datasets to be understudied across most registration literature. We plan to expand our focus on Abdominal data in future work.

---

> > ### Author Response · Authors · 2024-12-01
> > **Gentle reminder to respond to rebuttal**
> >
> > Dear Reviewer AqRJ,
> >
> > As the discussion period is wrapping up (in 2 days), we kindly ask that the reviewer reevaluate their score if all
> > their questions and concerns have been clarified.
> >
> > If there are still some concerns, please let us know so that we can respond accordingly.

---

### Official Review · Reviewer_bFGP · 2024-10-27

**Soundness:** 3
**Presentation:** 3
**Contribution:** 2
**Rating:** 5
**Confidence:** 4

**Summary:**

This paper presents a new deep learning-based multi-resolution image registration approach for medical data that deviates from the classical grid-like structure needs/approaches by incorporating principles for geometric learning. More specifically, it is proposed to use the Lagrangian reference frame (instead of an Eulerian one) and to model image features as nodes in an adjustable graph instead of on a static grid. It is claimed that this reduces resampling operations by moving into a continuous space for all computations. The proposed method is evaluated on synthetic brain deformations and additionally on three publicly available image registration benchmark datasets (2 brain, 1 lung). The experiments show that the proposed method achieves comparable or better results than the other learning-based registration methods used as baselines.

**Strengths:**

- I really appreciate the overall framing of the paper that challenges the common paradigms used in DL-based image registration and I believe that in the context of DL, this approach is novel and the paper is also easy to follow.
- Having a new method available that is less parameter intense is nice (although the memory footprint is concerning).
- Baselines include (some) relevant SOTA approaches and the chosen datasets cover two very common registration tasks.
- I like the framing (and use) of attention as a data-driven interpolation scheme and its integration into the framework.

**Weaknesses:**

- In general, I believe that the overall method is novel, but I am not sure that the chosen application scenarios actually showcase any tangible advantages. In my mind, such a Lagrangian frame-based setup makes most sense in a scenario where the input data is sparsely and irregularly sampled. However, this is not the case in the applications tackled in this paper. I would have expected to see something that either only involves shapes or point clouds to be registered or some artificial scenarios where only data at certain patches or key points is available (e.g., to mimic intra-interventional registration problem arising in radiation therapy). Given the presented applications, I fail to see the claimed benefits of the presented methods. However, in the area of such explicitly spare problems prior work on geometric deep learning already exists (e.g., Hansen et al.) as acknowledged by the authors, which then challenges a bit the novelty of this work.

- Additional problems that I see with the conducted experiments are that half of them operate on somewhat simple synthetic deformations and for the real-world scenarios (brain and lung registration) I miss any statistical tests that really back up the claimed improvements over the chosen baselines. While the baselines cover some of the SOTA approaches in DL-based image registration, it would have been nice to see additional specific, relevant baselines for the datasets chosen (e.g., LapIRNv2, which won last year's Learn2Reg challenge on the NLST data with a TRE of less than 1.5 mm). I am also missing traditional approaches such as ANTs as baselines as right now only a FFD/B-spline-based method is included.

- I also find the number of foldings in the real-world results rather high and concerning. Unfortunately this is not really discussed in the paper (or I have not been able to find it) and I would be interested in learning the reasons for this. Is this a result of the mostly implicit regularization employed here that is not strong enough? While I acknowledge that the authors are not explicitly focusing on presenting a diffeomorphic registration framework, this aspect warrants further investigation and improvements and/or a discussion on how this could be achieved.

- The presented method appears to be quite memory hungry, but the paper never makes this explicit. I would have expected to see some numbers here instead of the rather abstract discussion in Sec. 3.2.. Right now, it is not really possible to assess this crucial aspect of the paper.

**Questions:**

- Why do the authors think that the chosen application scenarios are actually helpful in showcasing the potential benefits of their method?
- Why were other potentially more competitive baselines not included in the comparison?
- How do the authors interpret the foldings in the real-work applications?
- What is the exact memory footprint of the method in comparison to the baselines for the chosen applications?

---

> ### Author Response · Authors · 2024-11-22
>
> We kindly thank the reviewer for the encouraging remarks and shared enthusiasm for our work.
>
> ## Experiment changes
>
> > T-tests
>
> Following the suggestions, we add paired t-tests between our method and each baseline on Table 2, marking lack of significance with an asterisk.
>
> > Additional baselines
>
> We recognize the relevance of the recommended baselines and thus extend our results section with the multi-resolution Symmetric diffeomorphic SyN algorithm of the ANTs framework [1] (an iterative optimization method), as well as LapIRN and FourierNet (learned methods used in the Learn2Reg challenge).
>
> ## Rebuttal
>
> >Geometric Deep Learning (GDL)
>
> We begin by addressing possible misconceptions related to GDL. Despite GDL predominantly associated with sparse data modalities, the purpose of the paradigm is to consolidate data manifolds with the underlying geometric structure which they rest upon. In deformable image registration (DIR), these would be equivalent to the grid of pixels/features and the Euclidean space in which this grid exists. We believe that explicitly respecting the coordinate system within an architecture is important across all registration modalities. While this is performed out of necessity in sparse data domains, we argue that it also offers an enticing formulation for grid-based DIR to reduce the black-box nature of current learned architectures.
>
> We would like to emphasize that this is a principle architectures like LapIRN do **not** follow. Their multi-resolution approach predicts the transformation residual at each subsequent resolution. However, each deformation is being predicted without full knowledge of the overall transformation up to that point. One could nonetheless argue that LapIRN does include skip-connections of features to the next resolution (prior to a given resolution’s final residual block), and as such *some* information is being passed onto finer resolutions about what kind of transformation their respective previous layer are likely to predict, but this information about the overall transformation is carried implicitly through the latent space. Our approach differentiates itself from current DIR works by representing the current transformation explicitly within the architecture.
>
>
> > Sparse data domains
>
> In cloud-point/surface registration literature (or approaches such as Hansen et al. that aim to sparsify the space by extracting ‘key-points’), graphs are constructed using closeness and similarity heuristics. Aspects such as dense supervision at any coordinate, pooling/upsampling operations, regularization, and neighborhood formulation are not trivial or efficient to formulate outside grid-based domains. The majority of our architecture follows standard grid-based designs, only employing graphs for the learned decoder operations (while nonetheless preserving grid-structure, see below). These, in simple terms, are a way to formulate the following ideas:
> - Tau: The regions of the source and target domains that should interact with one-another, should be informed by the current deformation.
> - Delta: The upsampling hierarchy should explicitly carry deformations across resolutions (as opposed to predicting deformations independently at each resolution).
> While we don’t doubt our contributions could be adopted in sparse data regimes, we believe this is well outside of the scope of our work.
>
>
> > Memory efficiency
>
> We would like to clarify how our method differs from geometric deep learning approaches on sparse data. In sparse domains, memory consumption is a notorious limitation due to the lack of structure in the data. Since our data is structured as a grid, we can instead map source coordinates into target grid indices to efficiently determine neighborhoods. We expand section 2.4 to include a paragraph on our memory-efficient implementations of Tau/Delta functions, where we discuss code-level optimization details such as the use of  Pytorch built-in grid unfolding operations to find nearest neighbors. We also include a table in Appendix D with VRAM consumption for each baseline, showing our memory consumption is in the range of SOTA baselines.
>
> > Regularization
>
> Our approach supervises deformations on a grid domain, hence we employ the same explicit regularization implementations that any other (grid-based) DIR approach would use (specifically ‘finite-differences’ in our case). In fact, since our approach explicitly models the coordinates of each source grid element within the architecture, we can regularize (and supervise) the deformation after each individual intermediate function at each resolution (via $coord_{current} - coord_{initial}$). While we chose our regularization hyperparameters based on qualitative displacement field visualizations, we agree with the reviewer that the reported folding metrics on Experiment 2 are (on 2 of the 3 experiments) higher than most baselines. This could be counteracted by simply increasing the regularization factor of the loss function.

---

> > ### Comment · Reviewer_bFGP · 2024-11-27
> >
> > I would like to thank the authors for the additional work they put into the paper to address my comments. However, my assessment of the paper remains basically the same. I actually believe that the additional experiments and the significance tests showcase that the presented method does not provide any tangible benefits over existing techniques. I still believe that the presented method would be well-suited for problems where the data is sparsely represented or where data does not live on a grid-like structure, but unfortunately, the paper does not showcase this.
> >
> > I am also a bit confused about the regularization. What is meant by using the same "explicit regularization implementations that any other (grid-based) DIR approach would use"? There is a huge difference between the more implicit regularization in many B-spline-based methods and more explicit regularizers (e.g., diffusion, curvature, elastic, learning-based prior, ...). I still believe that the number of foldings reported in the displacement fields is too large to be tolerable and I agree that more regularization (= imposing additional field smoothness) would reduce this, but most likely by negatively affecting accuracy. If the authors believe that this is not the case, then this would need to be shown.

---

> > > ### Author Response · Authors · 2024-12-01
> > > **Reply to Reviewer bFGP Part 1**
> > >
> > > > Clarification on regularization
> > >
> > > ## Explicit regularization
> > >
> > > The loss function we describe in Eq(1) follows the standard structure found in the DIR literature where a Dissimilarity function is minimized with an added weighted regularization cost :
> > > $$\mathcal{J}(T,S,\phi) = \mathcal{D}(T,S\circ\phi) + \lambda \mathcal{R}(\phi)$$
> > > The $\lambda$ hyperparameter is your standard regularization weight on the transformation $\phi$. In our case, the regularization consists of a bending energy [1] function to encourage smoothness on the transformation, which is implemented using finite differences of the predicted transformation grid $\phi$.
> > > More concretely, since our decoder architecture predicts a residual transformation at each $\tau$ and $\delta$ layer, we can obtain the total transformation and apply Eq(1) at each intermediate layer.
> > >
> > > ### Single resolution
> > >
> > > Let us assume we only work on a single resolution (no $\delta$ function). $\tau$ predicts a transformation residual $\Delta \phi_{current} = \tau(T, S, \phi_{current}) = \phi_i$, where $\phi_{current} =  \sum_{j=0}^{i} \phi_{j})$).
> > > After N iterations of $\tau$ the total transformation would be:
> > > $$\phi_{total} = \sum_{i=0}^{N-1} \phi_{i} = \sum_{i=0}^{N-1} \tau(T, S, \sum_{j=0}^{i} \phi_{j})$$
> > >
> > > We’ll use the notation $\tau_i = \tau(T, S, \sum_{j=0}^{i} \phi_{j})$ for the predicted residual at the $i$th iteration of $\tau$.
> > >
> > > The repeated application of the $\tau$ over N steps would thus be supervised using:
> > > $$\mathcal{J_{total}} = \mathcal{J}(T,S,\tau_{0}) + \mathcal{J}(T,S,\tau_{0} + \tau_{1}) + … + \mathcal{J}(T,S,\tau_{0} + … + \tau_{N-1}) $$
> > > $$= \sum_{i=0}^{N-1}\mathcal{J}(T,S,\sum_{j=0}^{i} \phi_{j})$$
> > > $$= \sum_{i=0}^{N-1}\mathcal{D}(T,S\circ \sum_{j=0}^{i} \phi_{j}) + \lambda \mathcal{R}(\sum_{j=0}^{i} \phi_{j}) $$
> > >
> > > In other words, our approach supervises dissimilarity and regularization of the current transformation after every $\tau$ layer.
> > > - In terms of dissimilarity: This encourages $\tau$ to predict the total transformation with as few iterations as possible.
> > > - In terms of regularization: This encourages $\tau$ to predict transformation residuals such that the total transformation up to that step preserves our space's structure.
> > >
> > > ### Multi-resolution
> > >
> > > In the multi-resolution setting, we want to extend the aforementioned loss function to additionally encourage the transformation at the coarser layer to provide a good initialization ($\phi_{start}$) for the next resolution of deformations. We use the superscript to represent a given resolution's grid dimensions. If an image has dimensions (H, W), the second-to-last resolution would have dimensions(H/2, W/2) (assuming pooling factor 2) and its total transformation would be $\phi_{total}^{(H/2,W/2)}$. The $\delta$ function would output the (H, W) resolution's starting transformation:
> > > $$\phi_{start}^{(H, W)} = \delta(S^{(H/2, W/2)}, S^{(H, W)}, \phi_{final}^{(H/2,W/2)})$$
> > >
> > > This is supervised using:
> > > $$\mathcal{J(\phi_{start}^{(H, W)})} = \mathcal{D}(T^{(H, W)},S^{(H, W)} \circ \phi_{start}^{(H, W)}) + \lambda \mathcal{R}( \phi_{start}^{(H, W)})  $$
> > >
> > > A given resolution's loss is thus the $\tau$ components mentioned above with an additional loss component for the initial transformation predicted by $\delta$:
> > > $$\mathcal{J_{total}^{(H, W)}} = \overbrace{\mathcal{J(\phi_{start}^{(H, W)})}}^{\delta} + \overbrace{\sum_{i=0}^{N-1}\mathcal{J}(T,S,\sum_{j=0}^{i} \phi_{j})}^{\tau}$$
> > >
> > > The total loss for all $L$ resolutions becomes the sum of each resolution's layers contributions:
> > > $$\mathcal{J_{total}} = \sum_{l=0}^{L-1} \mathcal{J_{total}^{(H/2^l, W/2^l)}}= \sum_{l=0}^{L-1} \left( \mathcal{J(\phi_{start}^{(H/2^l, W/2^l)})} + \sum_{i=0}^{N-1}\mathcal{J}(T,S,\phi_{start}^{(H/2^l, W/2^l)} + \sum_{j=0}^{i} \phi_{j}) \right)$$
> > >
> > > In summary, each intermediate layer's contribution can be supervised and **explicitly** regularized directly after its prediction. Additionally, since the architecture is end-to-end differentiable, a given layer also receives loss contributions from subsequent layers (including layers in further resolutions). This forces coarser resolutions to predict transformations, leading to finer resolutions to predict accurate and spatially well-behaved transformations.
> > >
> > > If the number of folds is too high, a user could increase the $\lambda$ parameter. Furthermore, our formulation allows for applying different regularization weights at different resolutions (e.g. If you want to enforce the coarsest resolution to model more rigid-like transformation components, you could set its regularization $\lambda$ high while leaving the $\lambda$ of finer resolutions low to encourage them to capture the deformable portion of the total transformation.
> > >
> > > ## Implicit regularization
> > > Besides using the explicit regularization term, our method is implicitly regularized by the multi-resolution approach. However, the usage of the learned attention mechanisms for both \tau and \delta do not offer any regularization properties.

---

> > > > ### Author Response · Authors · 2024-12-01
> > > > **Reply to Reviewer bFGP Part 2**
> > > >
> > > > > Well suitedness on sparse data and state-of-the-art performance
> > > >
> > > > We respectfully disagree with the reviewer on this point. While we don’t doubt some of our ideas could be useful in sparse domains, many aspects of our implementation are specific to grid-based data and would generalize quite poorly memory-wise (if not completely infeasible) to sparse domains (such as KNN and (dis)similarity computations).
> > > >
> > > > We still believe our main contribution of the presented method is not to demonstrate necessarily higher state-of-the-art performance in all datasets than all methods. On the contrary we wanted to emphasise that even though our work provides competitive SOTA results, the main contribution of this paper is to establish a theoretical foundation by which the architecture itself is explicitly modelling the transformations. Trying to avoid the current trend in deep learning registration that treats the networks as a black box, we designed our method to be explainable and to draw parallels between well-studied ideas of the field and modern deep learning architectural components, e.g. b-spline registration and attention mechanisms. Moreover, we envision that our method can pave the path for incorporating the spacing of the data in the learning process, which is being considered only in the iterative optimisation methods.

---

### Official Review · Reviewer_ZUxW · 2024-11-04

**Soundness:** 3
**Presentation:** 3
**Contribution:** 3
**Rating:** 6
**Confidence:** 4

**Summary:**

This paper presents a novel framework for deformable medical image registration that introduces a geometric deep learning-based approach, allowing the model to tackle large deformations and substantial geometric shifts. The key innovation lies in leveraging a Lagrangian framework, where deformations are applied without relying on grid constraints, which is a fresh approach in medical imaging registration. The model’s design also incorporates a multi-resolution framework, which progressively refines transformations from coarse to fine, reducing the need for intermediate resampling and preserving feature integrity. Experiments conducted on synthetic datasets demonstrate the model’s ability to outperform traditional grid-based methods, such as VoxelMorph, especially in handling large and complex deformations.

**Strengths:**

* Novelty in Approach: The use of a Lagrangian framework in the field of medical image registration is an innovative contribution. It allows transformations without grid dependency, addressing common limitations in traditional models.
* Ability to Handle Large Deformations: The model demonstrates strong performance in capturing large-scale geometric deformations, which many existing methods struggle with.
* Multi-Resolution Deformation Design: The progressive, coarse-to-fine deformation approach effectively reduces computational load at higher resolutions and ensures finer layers need only minimal adjustments, improving efficiency and accuracy.
* Strong Experimental Performance: The model achieves high accuracy in synthetic deformation tasks, showing its robustness in large-deformation registration scenarios, and reports low folding ratios, indicating better spatial regularity.

**Weaknesses:**

* Generalizability to Multi-Modal and Realistic Datasets: While the model performs well on synthetic datasets, it is unclear how it might adapt to multi-modal scenarios (e.g., T1-to-T2 MRI registration). There are limited details on how it can capture realistic anatomical variability beyond synthetic deformations. Expanding evaluation to multi-modal datasets with real-world challenges would strengthen the paper.
* Ambiguity in Experimental Design: In Table 1, the experimental setup lacks clarity. For instance, the distinction between the “Ours (feat. warp)” variant and the primary model (GeoReg) is not fully explained, leaving readers uncertain about the differences in architecture or training procedures between them.
* Sensitivity to Deformation Degree: The effect of deformation degree, such as the impact of the Brownian deformation parameter, on model performance is not well discussed. It would be helpful to understand how the model behaves under varying levels of deformation severity, as this might impact generalizability in diverse clinical cases.

**Questions:**

1. How does the model handle multi-modal registration beyond synthetic deformation? For example, would any special modifications be needed for realistic T1-to-T2 datasets?
1. Could you clarify the distinction between “Ours (feat. warp)” and the main GeoReg model in Table 1? How does the architecture or feature handling differ?
1. Does the degree of deformation in synthetic data (e.g., Brownian noise) influence model performance? Specifically, are there limits on the deformation magnitude where the model starts to fail?
1. Are there any practical limitations, such as longer training times or increased memory requirements, due to the Lagrangian framework compared to grid-based methods like VoxelMorph?

---

> ### Author Response · Authors · 2024-11-22
>
> We would like to thank the reviewer for the provided feedback and shared enthusiasm for our work.
>
> > ... it is unclear how it might adapt to multi-modal scenarios (e.g., T1-to-T2 MRI registration). There are limited details on how it can capture realistic anatomical variability beyond synthetic deformations.
>
> We acknowledge and agree with the reviewer’s concerns for the generalizability of the results presented in the synthetic deformations of Experiment 1. However, we would like to point out that Experiment 2 of our results section offers quantitative results on 3 real-world datasets, including a brain T1-to-T2 MRI dataset (the exact request of the reviewer), as well as a inhale-exhale lung CT dataset with large realistic deformations of the diaphragm region.
>
> Following the feedback of other reviewers, we have updated the wording of Experiment 1 on the revised manuscript to more explicitly reflect the proof-of-concept nature of the synthetic experiments in contrast to the real-world datasets already presented in Experiment 2.
>
> > ... the distinction between the “Ours (feat. warp)” variant and the primary model (GeoReg) is not fully explained.
>
> Although we offer a description on the difference between Ours (feat. warp) and (GeoReg) in Section 3.1 of the text, we believe that the reviewer raises a valid point regarding the clarity of this baseline. We have added clarification on the distinction in the caption of Tables 1 & 2.
>
> In short, in order to perform a traditional convolution, the feature elements need to lie on a standard grid. If we apply a deformation onto the feature grid, we can no longer naively apply a standard convolutional kernel over the grid. The 'feat. warp' baseline offers a Eulerian view over the feature domain: If a deformation is applied to the feature grid, we need to warp it back onto the standard grid positions before a convolution can be applied again. In the other hand, our main method (GeoReg) overcomes the need to perform resampling on the high-dimension feature grids by taking a Lagrangian view on the feature domains using the generalized interpretation of a convolution offered in the geometric deep learning paradigm, where relative coordinates are explicitly incorporated into the convolution operation.
>
> >  ... the impact of the Brownian deformation parameter, on model performance is not well discussed. It would be helpful to understand how the model behaves under varying levels of deformation severity.
>
> On the topic of Brownian deformation and its impact on model performance. We have added a section to Table 1 with no affine deformation component present where we examine the effect of three increasing levels of Brownian components.
> We would however like to make the reviewer aware of the compute requirements of the proposed baselines and experiment extensions discussed across all reviewers. As of the time of posting this comment, the extensions of the revised version of Table 1 currently only include results for our method, but we hope to progressively fill in baseline performances throughout the rebuttal period (or in the worst case scenario be fully filled by the camera ready deadline).

---

> > ### Author Response · Authors · 2024-12-01
> > **Gentle reminder to respond to rebuttal**
> >
> > Dear Reviewer ZUxW,
> >
> > As the discussion period is wrapping up (in 2 days), we kindly ask that the reviewer reevaluate their score if all their questions and concerns have been clarified.
> >
> > If there are still some concerns, please let us know so that we can respond accordingly.

---

### Official Review · Reviewer_RU9j · 2024-11-04

**Soundness:** 2
**Presentation:** 3
**Contribution:** 2
**Rating:** 3
**Confidence:** 3

**Summary:**

This paper proposed to approach deformable image registration by using graph neural networks. The authors claimed that using geometric deep learning method would let registration task get rid of grid constraints. Experiments showed that the proposed approach achieved state-of-the-art performance on brain MRI and lung CT datasets.

**Strengths:**

1. The presentation is clear and easy to follow.
2. The motivation of formulating image deformation registration as a graph learning problem sounds interesting.
3. Experiments explored large deformation setups on synthetic data as well as lung CT.

**Weaknesses:**

1. The methodology soundness is somewhat questionable. The authors have not mentioned how their proposed model would ensure the graph well-posedness when learning the deformations. Directly representing images in 3D grids to graph nodes could potentially mis-construct the folding/surface structures.
2. To represent registration as graph learning, the authors should compare and discuss the relations between their method and the large body of *point cloud* [1] and *surface* [2] registrations.
3. The comparisons are not sufficient/convincing and not up-to-date given the fast development of medical image registration field.
(1) Traditional physics-based models for diffeomorphic image registration, e.g., LDDMMs;
(2) Recent learning-based deformable registration models, e.g., SynthMorph, GradICON. For more models, authors could refer to Learn2Reg MICCAI2024 challenge (https://learn2reg.grand-challenge.org/).

[1] Shen et al. Accurate point cloud registration with robust optimal transport. NeurIPS, 2021.

[2] Hoopes et al. TopoFit: Rapid Reconstruction of Topologically-Correct Cortical Surfaces. PMLR, 2022.

[3] Vercauteren et al. Non-parametric Diffeomorphic Image Registration with the Demons Algorithm. MICCAI 2007.

[4] Hoffmann  et al. SynthMorph: learning contrast-invariant registration without acquired images. TMI, 2021.

[5] Tian et al. GradICON: Approximate diffeomorphisms via gradient inverse consistency. CVPR, 2023.

**Questions:**

My main questions are in the lack of discussion on related work and missing experimental comparisons. Please refer to weakness.

---

> ### Author Response · Authors · 2024-11-22
>
> We would like to thank the reviewer for the written feedback. However, we believe there exists a substantial misunderstanding of our methodology and contributions based on the comments provided in the Strengths and Weaknesses sections. We think the graph-based framing of our method may have led the reviewer to inaccurately believe our work closely relates to methods specializing on sparse data modalities. We hope this rebuttal may shed light on common misconceptions and pursues the reviewer to reevaluate their rating assessments.
>
> We would like to start by pointing out that despite geometric deep learning being predominantly associated with sparse data modalities, the purpose of the field is to consolidate data manifolds with the underlying geometric structure which they rest upon. In the DIR domain, these would be equivalent to the grid of pixels/features and the Euclidean space in which this grid exists. Our work’s contribution aims to make this relationship explicit in the context of DIR by building the geometric constraints of our coordinate system directly into the feature-processing architecture of the deformation process.
>
> Weakness 1 leads us to believe the reviewer is under the impression that our work refrains from using a grid structure altogether. **This is incorrect**. Our architecture assumes a 3D grid structure on the Source domain throughout the entire architecture, both in formulation and implementation. This is hardly different from similar multi-resolution DIR works:
> - A 3D convolutional encoder is used to extract feature grids from the images.
> - Features are used to predict displacements from the original grid and regularization is applied to maintain the surface structure.
> - The predicted displacement grids are densely supervised using trilinear interpolation of image values.
>
> Our contribution only sets itself apart in how information from the target domain is aggregated (Tau function) and how the deformation is carried across resolutions (Delta function).
>
> In simple terms, our graph operations are a way to formulate the following ideas:
> - Tau: The regions of the source and target domains that should interact with one-another, should be informed by the current deformation of the source domain.
> - Delta: The upsampling hierarchy should be set up so as to explicitly carry deformations across resolutions (as opposed to predicting deformations independently at each resolution).
>
> While the generalization of the convolution we offer in section 2.2 is one typically only found in graph neural network research, we would like to offer a more intuitive example from recent trends in Visual Transformer literature that aim to localize attention operations within grids. The Swin Transformer is such an example where attention is performed on varying sizes of local patch neighborhoods. The permutation-invariance property of the attention mechanism is best defined with set operation notation, but the overall architecture is nonetheless performed on a grid. In fact, the grid structure defines the neighborhoods (graph edges) and information on relative coordinates is positionally encoded into features.
> While our Delta function is similar in many ways to the Swin Transformer, our Tau function further extends this principle by being able to selectively utilize the neighborhood closest to the current world-coordinate of a given source “patch”.
>
> Furthermore, we would like to clarify upon the reviewer’s notion of “graph learning” mentioned in Strength 2 and Weakness 2, and the proposed connection to cloud-points/surfaces. The structure of our graphs is **not** learned. While much of the cloud-point registration literature aims to extract sparse key-point and construct graphs based on closeness and similarity heuristics, this is not how our approach works.
>
> We believe our ideas are best formalized in the domain of graphs (grids are a special type of graph afterall) so as to make the position of deformed grid nodes explicit. Additionally, aspects such as dense supervision at any location in space, pooling/upsampling operations, regularization, and neighborhood formulation are not trivial or efficient to formulate outside of grid-based domains. While we don’t doubt our contributions could be adopted in sparse data regimes, so could the contributions of any DIR work. We believe this is well outside of the scope of our work.
>
> Lastly, Weakness 3 proposes various baselines:
> - We agree with the reviewer on the relevance of LDDMM as a baseline. We update our experiments to include LDDMM[2].
>
> However, we are unsure how the following baselines are relevant to our method:
> - SynthMoprh proposes the use of synthetic data generation to improve performance, a line of work orthogonal to our architectural contributions. They use a VoxelMorph architecture, which we already include in our baselines.
> - Similarly, GradICON and ICON are works on regularizers atop of standard architectures. Their experiments also use a Unet.

---

> > ### Author Response · Authors · 2024-12-01
> > **Gentle reminder to respond to rebuttal**
> >
> > Dear Reviewer RU9j,
> >
> > As the discussion period is wrapping up (in 2 days), we kindly ask that the reviewer reevaluate their score if all their questions and concerns have been clarified.
> >
> > If there are still some concerns, please let us know so that we can respond accordingly.

---

### Author Response · Authors · 2024-11-22

We would like to thank all reviewers for the shared enthusiasm regarding the novelty of our contributions. We appreciate the constructive feedback regarding the proposed improvements of our results section. We outline the extensions to baselines and experiments below.

We hope to invite all reviewers to engage in the author/reviewer discussions and kindly encourage them to revisit their scores if they feel their concerns have been addressed. We are readily available to respond to any further questions and concerns. We refer reviewers to the revised manuscript where all changes are highlighted in red.

We have received a number of concerns regarding our approach’s ability to model the spatial structure of our images and apply regularization. We believe these concerns stem from an assumption that geometric deep learning (GDL) is exclusively associated with sparse data modalities. The GDL paradigm offers a pathway to incorporate information of the underlying coordinate system into the feature processing mechanisms of neural networks. While this is necessary for sparse modalities, the principle generalizes to grids. After all, grid-based convolutional neural networks are a special case of graph neural networks where relative coordinates between graph nodes are implicitly built into the learned kernels.

Our work is, at its core, a grid-processing architecture like common learning-based deformable image registration (DIR) architectures. We use a 3D convolutional encoder, densely supervised a pixel-wise grid of deformations, regularize using standard finite-difference functions, etc., in the same ways most of our baselines operate. As such, the topological guarantees offered by our approach hardly differ from concurrent works.

Similarly, while we formulate our decoder layers using GDL notation, their implementation adapts these concepts to grid-based neighborhoods and operations. Following concerns on memory usage of our method, we have added a table displaying VRAM usage for all approaches to Appendix D, showing our approach falls in the same range as other SOTA architectures. We also extend section 2.4 with code-level implementation details for the memory-efficient formulation of our layers in grid domains.

We believe that, even though our work provides competitive results with SOTA, the main contribution of our manuscript lies in establishing a theoretical foundation by which transformations can be explicitly modeled within a deep learning architecture. We think that this contribution opens up research avenues to reduce the black-box nature of current learned DIR paradigms, incorporate ideas from conventional image registration into deep learning architectures, and tackle known issues such as data anisotropicity.

We hope that by having extended our results with the experiments suggested by the reviewers, our manuscript’s claims may become more convincing in the eyes of new readers.

## Further Experiments and Baselines

### Experiment 2

Following the reviewers’ suggestions, we proceed by running more baselines for the real-world deformable registration experiment, as well as performing statistical t-tests (p < 0.05) between our proposed method.
More specifically, we incorporate two iterative optimization based methods, namely the symmetric diffeomorphic algorithm (SyN) of the widely used ANTs framework [1] and a Large Deformation Diffeomorphic Metric Mapping (LDDMM) [2] algorithm implemented in PyTorch [https://github.com/brianlee324/torch-lddmm]. Moreover, regarding learning-based methods, we include two baselines used in the Learn2Reg challenge: LapIRN [3] that tackles large deformations in a coarse-to-fine manner using CNNs and FourierNet [4].

### Experiment 1

In response to the reviewers’ suggestions we expand Table 1 with two sections: one where we remove the affine component of the synthetic transformation and we gradually increase the Brownian local deformation component, and a table where we try to recover a full rigid transformation combining (rotation + translation + scaling).  However, due to the large amount of experiments proposed, the current state of our draft provides partial results for some of the baselines. We would like reviewers to note that we plan to finish filling in these results by the end of the rebuttal period.

[1] Avants et al. Symmetric diffeomorphic image registration with cross-correlation: evaluating automated labeling of elderly and neurodegenerative brain. Medical image analysis. 2008.

[2] Beg et al. Computing large deformation metric mappings via geodesic flows of diffeomorphisms. International journal of computer vision. 2005.

[3] Mok and Chung. Large deformation diffeomorphic image registration with laplacian pyramid networks. MICCAI. 2020.

[4] Jia X et al. Fourier-net: Fast image registration with band-limited deformation. AAAI. 2023.

---

### Meta-Review · Area_Chair_rCx4 · 2024-12-19

**Metareview:**

This paper presents an new approach for deformable image registration of MR and CT images. While the introduction of geometric deep learning into deformable registration seems to be new, the reviewers raised some concerns on the soundness, experiments and etc. I also have some concerns on the results. For example, the results in Figure 3 is not technically sound such as the results from Voxelmorph and Transmorph. These approach normally would adopt a two-step approach where the first step obtains a coarse registration and second step for finer results. But the results shown by the authors suggest that these approach work poor than affine transform. In addition, the visualization is from simulated results. I suggest the authors to show a real-world case to demonstrate.
Besides the above concerns, I also feel the title inappropriate as image registration covers both deformable and non-deformable registration while this method mainly deals with deformable one.

**Additional Comments On Reviewer Discussion:**

NA

---

### Decision · Program_Chairs · 2025-01-22

Reject